# Compromised transcription-mRNA export factor THOC2 causes R-loop accumulation, DNA damage and adverse neurodevelopment

Rudrarup Bhattacharjee [1,2], Lachlan A. Jolly[2,3], Mark A. Corbett [1,2],
Ing Chee Wee [4], Sushma R. Rao[1,5], Alison E. Gardner[1,2], Tarin Ritchie [1,2],
Eline J. H. van Hugte[6], Ummi Ciptasari [6], Sandra Piltz [2,3,7],
Jacqueline E. Noll [8], Nazzmer Nazri [1,9], Clare L. van Eyk [1,2], Melissa White[2,3,7],
Dani Fornarino[1,2], Cathryn Poulton[10], Gareth Baynam [10,11,12],
Lyndsey E. Collins-Praino[4], Marten F. Snel[1,5], Nael Nadif Kasri[6], Kim M. Hemsley[1,9],
Paul Q. Thomas[2,3,7], Raman Kumar [1,2,13] & Jozef Gecz [1,2,13] ✉

We implicated the X-chromosome *THOC2* gene, which encodes the largest subunit of the highly-conserved TREX (Transcription-Export) complex, in a clinically complex neurodevelopmental disorder with intellectual disability as the core phenotype. To study the molecular pathology of this essential eukaryotic gene, we generated a mouse model based on a hypomorphic *Thoc2* exon 37–38 deletion variant of a patient with ID, speech delay, hypotonia, and microcephaly. The *Thoc2* exon 37–38 deletion male (*Thoc2^{Δ/Y}*) mice recapitulate the core phenotypes of *THOC2* syndrome including smaller size and weight, and significant deficits in spatial learning, working memory and sensorimotor functions. The *Thoc2^{Δ/Y}* mouse brain development is significantly impacted by compromised THOC2/TREX function resulting in R-loop accumulation, DNA damage and consequent cell death. Overall, we suggest that perturbed R-loop homeostasis, in stem cells and/or differentiated cells in mice and the patient, and DNA damage-associated functional alterations are at the root of *THOC2* syndrome.

Human neurodevelopment is a highly orchestrated and complex process. The identification of naturally occurring gene variants leading to neurodevelopmental disorders (NDD) highlights the involvement of over 2300 different genes (https://www.omim.org)[1] and a myriad of cellular, molecular, and developmental mechanisms, only a few of which are being successfully targeted with precision therapies[2]. Many essential biological processes are compromised by such NDD gene variants and often knockout of these genes cause embryonic lethality. One such example is variants in the X-chromosome gene *THOC2* that cause NDDs with clinically heterogeneous presentations, now we call

*THOC2* syndrome[3–5]. *THOC2*, a highly constrained gene, encodes the largest subunit of the THO subcomplex in the highly-conserved Transcription-Export (TREX) complex. The TREX complex is essential for transcription, mRNA processing and export, preventing DNA damage, and maintaining ESC self-renewal, pluripotency and differentiation during embryogenesis[6,7].

The TREX complex is composed of a THO sub-complex (THOC1-3, 5–7) and eight accessory proteins (UAP56/DDX39B, ALYREF/THOC4, UIF, CHTOP/SRAG, CIP29/SARNP, POLDIP3, ZC3H11A and LUZP4)[7–9]. THOC2 is a scaffold protein critical for TREX assembly and function[10,11].

The functional TREX is co-transcriptionally loaded onto the mRNAs and transfers the mature transcripts to the export adapters NXF1-NXT1 via ALYREF, eventually exporting the mRNAs to cytoplasm through the nuclear pores[7,12,13]. In addition to its mRNA export function, the TREX complex has emerged as a multifunctional molecular machine with critical roles in transcription, 3' mRNA processing, stress responses, mitotic progression, and genome stability[7,14–20].

Many studies in lower organisms have illuminated the essential roles of *THOC2* in cellular and developmental processes. For example, yeast lacking *Tho2* (mammalian *THOC2* homolog) were cell growth inhibited[10]. *C. elegans thoc2* knockouts were slow-growing, sterile, had defects in specific sensory neurons and died prematurely[18]. In *Drosophila*, *THO2* depletion in S2 cells inhibited export of mRNA encoding heat shock protein (Hsp) and cell proliferation[21]. Zebrafish *Thoc2* is essential for embryonic development[22]. In mice, studies in embryonic stem cells (mESC) show *Thoc2* and *Thoc5* regulate self-renewal by selectively binding and regulating the export of mRNAs encoding the Yamanaka factors (*Oct3/4*, *Sox2*, *Klf4* and *c-Myc*)[6]. However, in vivo *Thoc2* studies in mammalian models have been non-existent likely because of apparent embryonic lethality of knockout lines (International Mouse Phenotyping Consortium (IMPC), http://www.mousephenotype.org)[23]. Indeed, complete *THOC2* depletion causes severe nuclear mRNA retention in mammalian cells[24]. Whilst these data highlight the essential role of *THOC2* in cell and organism function, its involvement in human development and disease has also expanded. For example, a de novo translocation resulting in a *PTK2-THOC2* gene fusion with *THOC2* expression knockdown was implicated in cognitive impairment and cerebellar hypoplasia[25]. Cytoplasmic aggregation of THOC2/THO proteins impairs nucleo-cytoplasmic transport and may be a common process explaining the pathology of neurodegenerative diseases, including Huntington's, Alzheimer's and amyotrophic lateral sclerosis[26]. These findings, along with published[3–5] missense, microdeletion or splicing-defective *THOC2* NDD variants suggest that *THOC2* performs multiple critical roles with major impacts on neuronal function. We premised that the brain is sensitive to THOC2 function given its heightened requirements for and diversity of transcription and protein synthesis. However, the role of *THOC2* and TREX complex in normal brain development and, importantly, in causing NDDs remains elusive largely due to our inability to create relevant in vivo models of this essential gene. So, the broader functional understanding of *THOC2* in mammalian development remains unknown. Thus, we sought to generate a preclinical mouse model to delve deeper into the unresolved aspects of *THOC2* biology with a view to understand the molecular and cellular roles in brain development as well as mechanism of the disease pathology in individuals with *THOC2* pathogenic variants.

We reasoned that naturally occurring hypomorphic (i.e., partial loss of function) variants in *THOC2* patients with NDDs and thus compatible with life offer the opportunity for in vivo investigations by overcoming lethality. We utilized one such hypomorphic *THOC2* variant (NC_000023.11:g.123609724_123612089del, ClinVar: VCV000804360.1) we reported in a patient with intellectual disability (ID), speech problems, short stature, low body weight, sensorineural hearing loss and microcephaly[5] to generate a mouse model with a microdeletion encompassing Exons 37 and 38. We show that *Thoc2* is critical during early post-fertilisation development, but mice can survive into adulthood. Neurobehavioral testing shows that the *Thoc2* exon 37–38 deletion male mice (called *Thoc2*[Δ/Y]) have learning, memory, and sensorimotor deficits, recapitulating phenotypes observed in most of the patients with *THOC2* syndrome. Our transcriptomic and proteomic data shows major dysregulation in pathways associated with transcription, splicing and higher order cognitive functions. We show *Thoc2*[Δ/Y] mice have a reduced neural stem cell (NSC) pool, altered brain cytoarchitecture, and multiple impairments of neuron structure and function that lead to significantly compromised electrophysiological properties and synapse formation. However, to our surprise, we observe that mRNA export is not overtly affected in the *Thoc2*[Δ/Y] mice, but instead results in accumulation of R-loops and concomitant DNA damage. This mechanism is also evident in *THOC2* exon 37–38 deletion patient-derived dermal fibroblasts. Together, our data reveal a mechanism of *THOC2* NDD originating early in development and affecting cell's ability to resolve R-loops and consequently preventing DNA damage and causing cell death.

## Results

*THOC2* gene essentiality in eukaryotes[27], and absence of a knockout mammalian model suggest that its complete loss is incompatible with life[23]. However, we have previously reported 22 *THOC2* (19 missense, two splicing defective and one microdeletion) variants, 12 of which are hypomorphic and functionally validated in individuals with intellectual disability (ID) and variable expressivity of other traits including hypotonia, growth restriction and microcephaly[3–5]. We postulated that these naturally hypomorphic *THOC2* alleles could be utilized for generating a viable mouse model to gain insights into the THOC2/TREX biology as well as understanding the pathology of *THOC2* syndrome.

### Generation of a viable mouse model using a hypomorphic *Thoc2* exon 37-38 deletion variant

Our initial attempts to generate knock-in mice carrying either the p.Leu438Pro or p.Ile800Thr *THOC2* variant, that reduce the stability of THOC2 protein (and other THO subunits) in patient-derived cell lines[3], failed to produce any live offspring (see Supplementary Data 1). Subsequently, we generated a mouse model based on a pathogenic deletion that includes exons 37 and 38 that code for the extreme COOH-terminus of the THOC2 protein using CRISPR/Cas9 editing (Fig. 1a). Four founder heterozygous females (Δ/+) and two hemizygous males with exon 37-38 deletion (Δ/Y) were recovered (Supplementary Fig. 1a, b). We confirmed the deletion of genomic sequences coding for *Thoc2* exon 37-38 and absence of any off-target CRISPR events by whole genome sequencing. We further eliminated the off-target events by backcrossing the *Thoc2*[Δ/+] with WT C57BL/6JArc male mice for three generations. We confirmed the deletion of exon 37-38 sequences from the *Thoc2* mRNA in 30 days old *Thoc2*[Δ/Y] mice brain using RT-PCR and Sanger sequencing (Fig. 1b). Exon 37 and 38 deletion in the *Thoc2* mRNA is in-frame, not susceptible to non-sense mediated decay (NMD) and predicted to translate into a C-terminally truncated ~172 kDa compared to normal ~175 kDa THOC2 protein (Supplementary Fig. 1c). We analysed the adult *Thoc2*[Δ/Y] mice brain lysates by western blotting (WB) with two antibodies capable of distinguishing the full length and the truncated THOC2 proteins (Fig. 1c). As expected, we observed a slightly smaller THOC2 protein in the *Thoc2*[Δ/Y] compared to the full-length protein in the *Thoc2*[+/Y] mouse brains. The levels of truncated THOC2 protein were increased in *Thoc2*[Δ/Y] mice brains (Fig. 1c) as also observed in the *THOC2* exon 37-38 deletion patient fibroblasts and lymphoblastoid cell line (LCL)[5]. The *Thoc2*[Δ/+] female mouse brains also showed increased level of THOC2 protein (Supplementary Fig. 1d, lane 2). As expected, only the truncated THOC2 protein was present in the *Thoc2*[Δ/Δ] female mouse brains (Supplementary Fig. 1d, lane 3).

While breeding animals for experimental work, we recovered ~15–16% *Thoc2*[Δ/Y] animals (both embryos and postnatal) compared to the expected Mendelian genotypic ratio of 25% (Fig. 1d). We predicted reduced recovery of *Thoc2*[Δ/Y] mice was likely due to early post fertilisation death. We investigated this by performing in vitro fertilization (IVF) with *Thoc2*[Δ/+] mouse oocytes and *Thoc2*[+/Y] sperms and scoring the blastocysts' for *Thoc2*[+/Y] and *Thoc2*[Δ/Y] genotypes, using our rapid gDNA extraction-multiplex PCR method, 4 days after the IVF procedure (Fig. 1e). We found that only ~14% blastocysts were *Thoc2*[Δ/Y] (Fig. 1e), indicating that reduced recovery of the *Thoc2*[Δ/Y] mice was due to the death of zygotes that are much before the blastocyst stage. *THOC2* has higher expression at the blastocyst stage[6]. A recent report showed that THO subunits have higher expression during embryonic stages (E10.5

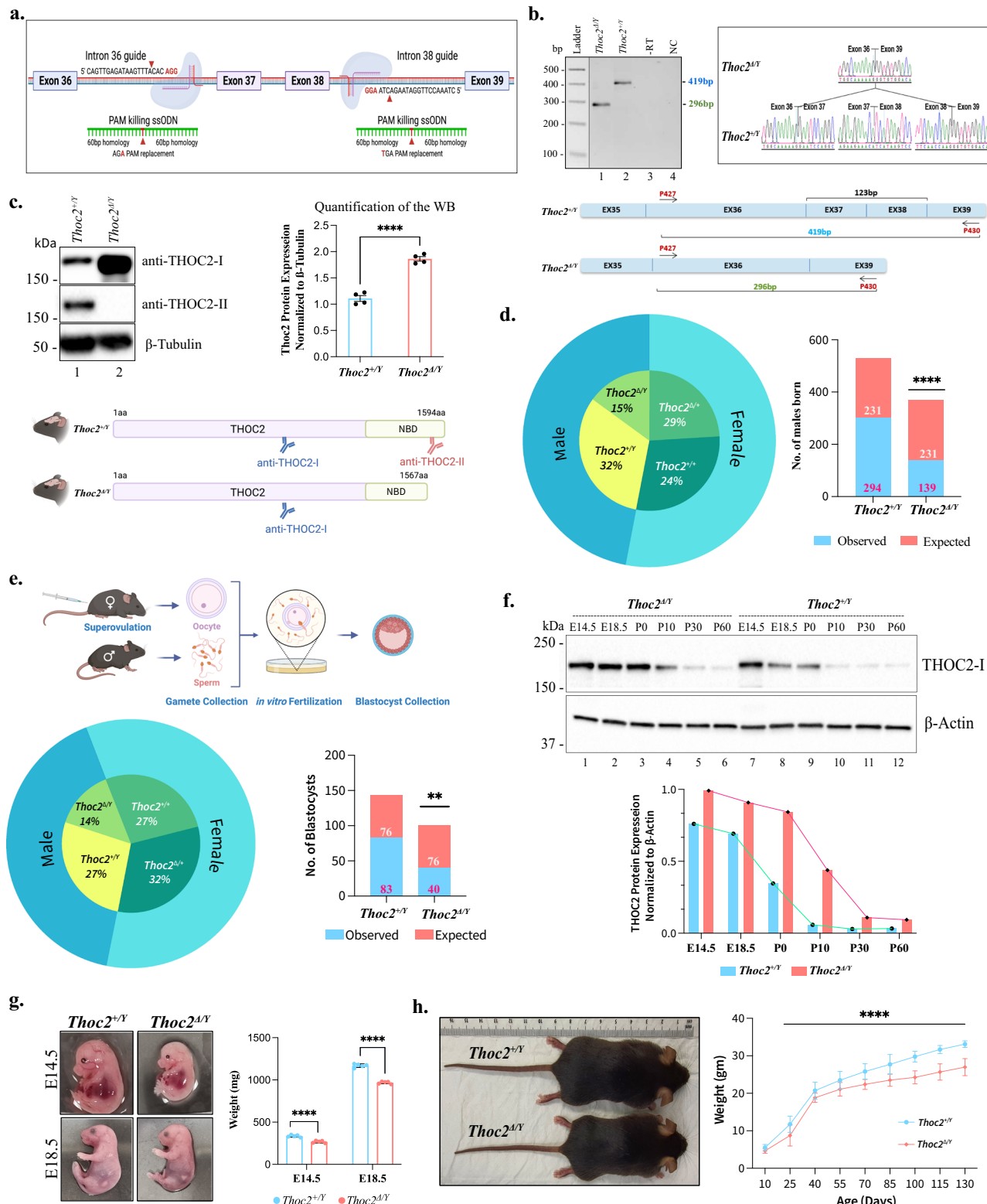

to E18.5) that declines sharply from postnatal day 3 (P3) onwards[28]. We asked if THOC2 and other TREX proteins also show a similar trend and analysed the levels of THOC2 protein in the brains of *Thoc2+/Y* and *Thoc2Δ/Y* mice at embryonic (E14.5 and E18.5) and postnatal (P0, P10, P30 and P60) stages. We found that the THOC2 protein levels are high at the embryonic stages and decline sharply from postnatal day 10 onwards (Fig. 1f). This developmental decline of THOC2 and other TREX subunit proteins in the brains of *Thoc2+/Y* and *Thoc2Δ/Y* mice (Supplementary Fig. 1e) agreed with the publicly available[28] and our

E14.5, E18.5 and P10 mouse brain RNA-seq data (Supplementary Fig. 1f). These results support a crucial role for *Thoc2* in early development, mostly post fertilisation and at a stage of pluripotent embryonic stem cells.

We observed that the *Thoc2Δ/Y* mice (embryo and adult) have significantly reduced weight (-15%) and smaller size compared to their *Thoc2+/Y* littermates (Fig. 1g, h), a phenotype similar to the patient with the *THOC2* exon 37-38 deletion variant (see Supplementary Data 2). The vital organs (e.g., brain, heart, lung, kidney, and liver) of the *Thoc2Δ/Y*

**Fig. 1 | Generation of *Thoc2* exon 37-38 deletion mouse model. a** Schematic diagram of CRISPR/Cas9 gene editing strategy for deleting exon 37 and 38 of the mouse *Thoc2* gene. **b** Reverse transcription-PCR DNA of *Thoc2*$^{+/Y}$ (419 bp) and *Thoc2*$^{Δ/Y}$ (296 bp) brain RNAs (left panel) with representative image of Sanger sequencing traces (right panel) showing deletion of exon 37-38 coding sequence in the *Thoc2*$^{Δ/Y}$ mRNA. Black line indicates that the lanes between the marker and PCR samples were deleted to generate this figure ($n = 4$ independent experiments). **c** Western blot showing THOC2 protein in *Thoc2*$^{+/Y}$ and *Thoc2*$^{Δ/Y}$ mouse brains. The schematic shows binding location of the two antibodies. β-Tubulin was used as loading control. Quantification of THOC2 protein levels in *Thoc2*$^{+/Y}$ and *Thoc2*$^{Δ/Y}$ mouse brains are also shown ($n = 4$ experiments); ****$p = 0.00004$; two-tailed unpaired student's *t* test. **d** Pie chart showing the percentage of different genotypes born in the *Thoc2* mouse colony. Observed and expected number of *Thoc2*$^{+/Y}$ and *Thoc2*$^{Δ/Y}$ mice born in the colony based on the monogenic pattern of Mendelian inheritance. ****$p < 0.0001$; $χ^2$ test. **e** Pie chart showing the percentage of different genotypes among the blastocysts analysed. Observed and expected number of *Thoc2*$^{+/Y}$ and *Thoc2*$^{Δ/Y}$ blastocysts are graphed. **$p = 0.0081$; $χ^2$ test. A schematic presentation of the in vitro fertilization experimental approach is shown. **f** Western blot and a graph showing THOC2 protein in *Thoc2*$^{+/Y}$ and *Thoc2*$^{Δ/Y}$ mouse brains at different embryonic (E14.5, E18.5) and post-natal (P0, P10, P30, P60) stages. β-Actin was used as loading control. THOC2 protein levels normalized to β-Actin were graphed ($n = 3$ independent experiments). **g** *Thoc2*$^{+/Y}$ and *Thoc2*$^{Δ/Y}$ E14.5 and E18.5 embryos ($n = 5$ embryos per genotype, per embryonic stage) and **h** *Thoc2*$^{+/Y}$ and *Thoc2*$^{Δ/Y}$ adult mice ($n = 50$ *Thoc2*$^{+/Y}$ and $n = 42$ *Thoc2*$^{Δ/Y}$ mice); ****$p < 0.0001$; two-way ANOVA, Bonferroni's multiple comparison test. All the data (except for **d**, **e**) presented are mean values ± SEM. Source data are provided as a Source Data file. Schematics in figures **a**, **c**, **e** were created with BioRender.com.

mice were smaller and lighter as compared to the *Thoc2*$^{+/Y}$ mice, suggesting a proportional global reduction in their body size and weight (Supplementary Fig. 1g). The P30 *Thoc2*$^{Δ/Y}$ mice showed a distinct shorter and flatter snout compared to the *Thoc2*$^{+/Y}$ mice (Supplementary Fig. 1h). Finally, we did not observe difference in the weight of 30–120 days old *Thoc2*$^{+/+}$ and *Thoc2*$^{Δ/+}$ female mice (Supplementary Fig. 1i).

Together, our data showed that we have generated a viable *Thoc2* exon 37-38 deletion mouse model that recapitulated smaller body size, presence of exon 37-38 deleted mRNA and C-terminally truncated protein as also observed in the *THOC2* exon 37-38 deletion patient[5].

### *Thoc2*$^{Δ/Y}$ mice have learning, memory and sensorimotor deficits

As the mouse model is based on a *THOC2* pathogenic NDD variant, we subjected eight-week-old *Thoc2*$^{Δ/Y}$ mice and *Thoc2*$^{+/Y}$ littermates to a battery of neurobehavioral tests to assess cognition, motor coordination, sensorimotor function, gross and fine motor skill, and anxiety-like behaviour (Fig. 2 and Supplementary Fig. 2).

**Cognitive function assessment.** We utilized the Morris Water Maze[29] and Barnes Maze[30] to assess spatial learning and memory and the Y-maze test[31] to assess spatial working and reference memory. The Morris Water Maze results indicated significant learning deficits in *Thoc2*$^{Δ/Y}$ mice, as these mice took longer time to locate the escape platform compared to their *Thoc2*$^{+/Y}$ littermates, even after 5 days of "learning trials" (Fig. 2a). The *Thoc2*$^{Δ/Y}$ mice also showed significantly fewer visits to the target quadrant and escape platform and spent much less time in the target quadrant or on the escape platform compared to their *Thoc2*$^{+/Y}$ littermates, suggesting that they had never fully learned the task (Supplementary Fig. 2a). Consistent with these results, *Thoc2*$^{Δ/Y}$ mice also showed significantly higher escape latencies than the *Thoc2*$^{+/Y}$ mice on both the second and third day of "learning trials" on the Barnes Maze, as well as on probe trials (Fig. 2b). On probe trials, *Thoc2*$^{Δ/Y}$ mice also made fewer visits to the old escape box location and spent much less time in the quadrant of the old escape box, providing additional support for the presence of spatial learning deficits in *Thoc2*$^{Δ/Y}$ mice (Supplementary Fig. 2b). Interestingly, however, both the *Thoc2*$^{+/Y}$ and *Thoc2*$^{Δ/Y}$ mice showed comparable performance in finding the new escape box, indicating cognitive flexibility is intact in *Thoc2*$^{Δ/Y}$ mice (Supplementary Fig. 2b). Given the significant learning deficits in *Thoc2*$^{Δ/Y}$ mice, we chose to further probe spatial working and reference memory using the Y-maze task. *Thoc2*$^{Δ/Y}$ mice showed a significantly reduced percentage of alternations between the maze arms compared to their *Thoc2*$^{+/Y}$ littermates (Fig. 2c), indicating an impairment of working memory.

**Motor function and sensorimotor control assessment.** The motor coordination and gait analysis from Rotarod[32] and Digigait[33] tests, respectively, did not show any difference between the *Thoc2*$^{Δ/Y}$ and *Thoc2*$^{+/Y}$ mice, suggesting no impairment in either gross motor coordination or overall gait (i.e., swing time, brake time) (Supplementary Fig. 2c, d). It is important to note, however, that *Thoc2*$^{Δ/Y}$ mice showed a significant decrease in stride length in the hind-paws, as well as a concomitant decrease in propel time of the forepaws (Supplementary Fig. 2d), which could indicate gait asymmetries. Interestingly, in the open field test[34], *Thoc2*$^{Δ/Y}$ mice travelled a significantly greater total distance as compared to their *Thoc2*$^{+/Y}$ littermates, indicating a potentially hyperactive locomotor phenotype in the *Thoc2*$^{Δ/Y}$ mice (Fig. 2d). Similarly, the *Thoc2*$^{Δ/Y}$ mice showed an increased distance travelled compared to their *Thoc2*$^{+/Y}$ littermates during both the elevated plus maze task[34] (Supplementary Fig. 2e) and the light-dark test[35] (Supplementary Fig. 2f). *Thoc2*$^{Δ/Y}$ mice also had higher number of entries into all arms on the elevated plus maze task (Supplementary Fig. 2e) and made a greater number of transitions in the light-dark test (Supplementary Fig. 2f). However, there was no difference between the groups for the time spent in either the outer or inner zone of the open field during task performance (Fig. 2d).

Next, we tested fine motor coordination in *Thoc2*$^{Δ/Y}$ mice using the beam walking test[36]. We found that the *Thoc2*$^{Δ/Y}$ mice showed a significantly higher number of foot slips while crossing the beam, suggesting that these mice had more difficulty in maintaining their balance and fine motor control during task performance (Fig. 2e). We then probed fine motor control using the pasta handling task[37]. Interestingly, the *Thoc2*$^{Δ/Y}$ mice took significantly longer time to consume one piece of pasta as compared to *Thoc2*$^{+/Y}$ littermates (Fig. 2f). This was associated with more frequent drops of the pasta, as well as more frequent adjustments to grip when holding and eating the pasta, collectively considered as 'atypical behaviours' (Fig. 2f). Similarly, *Thoc2*$^{Δ/Y}$ mice showed significantly increased time to contact and remove the tape when compared to their *Thoc2*$^{+/Y}$ littermates in the adhesive removal test[38] (Fig. 2g). Importantly, however, this was not associated with gross changes in grip strength (Supplementary Fig. 2g). Taken together, these results suggest that impairments observed are more indicative of deficits in fine motor control and sensorimotor function than skeletal muscle alterations.

**Anxiety assessment.** Surprisingly, on the elevated plus maze[34], the *Thoc2*$^{Δ/Y}$ mice showed a decrease in anxiety-like behaviour, as evidenced by increased time spent in open arm and centre of the maze and more frequent entries into the open arm, with a concomitant decrease in time spent in the closed arm (Supplementary Fig. 2e). Corroborating these observations, the *Thoc2*$^{Δ/Y}$ mice also spent more time in the light compartment and less time in the dark compartment during the light-dark test and took less time to emerge from the dark chamber into the light chamber, as compared to their *Thoc2*$^{+/Y}$ littermates (Supplementary Fig. 2f).

Taken together, the neurobehavioral studies indicate impairments in hippocampus-dependent spatial learning and memory, deficits in fine motor control and sensorimotor function, hyperactivity, and reduced anxiety in *Thoc2*$^{Δ/Y}$ mice, phenotypes that are similar to the *THOC2* exon 37-38 deletion patient (Supplementary Fig. 2h).

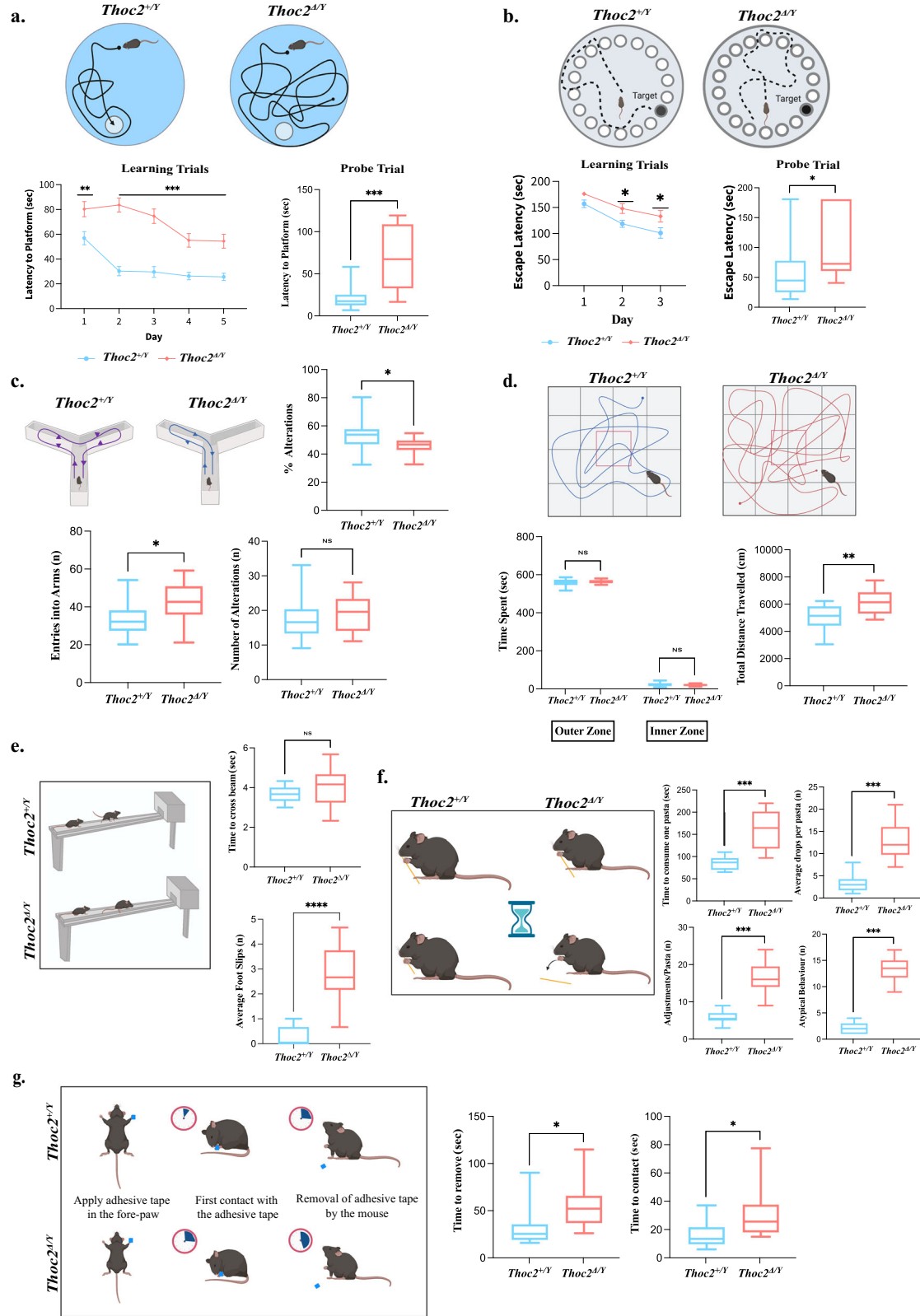

## Thoc2$^{\Delta/Y}$ embryonic and adult mice brain show significant dysregulation in genes associated with transcription, cell cycle, cell death and cognitive function

As the TREX/THO plays a conserved and central role in the process of transcription and subsequent export of mRNAs[7,15,19], we asked if the truncated THOC2 protein would cause specific transcriptional and/or mRNA export perturbations. Surprisingly, we did not observe any overt nuclear retention of mRNAs in Thoc2$^{\Delta/Y}$ cultured primary neurons as well as THOC2 exon 37-38 deletion patient fibroblasts (Supplementary Fig. 3a). We investigated if there was any transcriptomic dysregulation by performing RNA-seq of embryonic (E14.5, E18.5) and post-natal day 10 (P10) Thoc2$^{\Delta/Y}$ and Thoc2$^{+/Y}$ mouse brain total RNAs (specifically cortex and hippocampus; functionally compromised regions as indicated by the behavioural outcomes). We observed

**Fig. 2 | *Thoc2^{Δ/Y}* mice display multiple neurobehavioral deficits. a** Morris Water Maze test for measuring spatial learning and reference memory. Graphs showing latency to find escape platform over 5 days of learning trial (left) and the time taken to find the escape platform on the probe day (right); **$p = 0.0050$; ***$p = 0.0003$. **b** Barnes Maze test for assessing learning and memory function. Graphs showing latency to escape into the box over three days of trial (left) and time taken to escape into the box on the probe day (right) ($n = 13$ mice per genotype); *$p = 0.02$. For **a**, **b** two-way ANOVA, Bonferroni's multiple comparison test on the trial day graphs and two-tailed unpaired student's *t* test on probe day graphs. **c** Y-Maze spontaneous alteration test for assessing working memory. Graphs showing percentage of alterations between novel arm and previously visited arm, total number of arm entries and total number of alterations; *$p = 0.047$; unpaired two-tailed student's *t* test. **d** Open Field test assessing the general locomotor activity. Graphs showing time spent in different zones of the field (left) and total distance travelled in cm (right); two-way ANOVA, Bonferroni's multiple comparison test and **$p = 0.006$; two-tailed unpaired student's *t* test. **e** Beam walking test for assessing fine motor coordination. Graphs showing time taken by the mice to cross the beam (left) and number of foot slips (right); ****$p < 0.0001$. **f** Pasta handling task to assess fine motor-control. Graphs showing time taken to consume one pasta (upper left), number of times pasta has been dropped while eating (upper right), number of times mice adjusted their grip of the pasta (lower left) and number of atypical behaviours (lower right); ***$p = 0.0004$. **g** Adhesive removal test for assessing sensorimotor function. Graphs showing time taken by mice to contact the adhesive attached to its paw (upper) and to remove the adhesive from paw after contacting it (lower); *$p = 0.012$. For **e**–**g** unpaired two-tailed student's *t* test. Except **b**, all experiments were performed with $n = 14$ mice per genotype and the data presented are mean values ± SEM. NS non-significant. Source data are provided as a Source Data file. Schematics in figures **a**–**g** were created with BioRender.com.

significant transcriptional dysregulation in *Thoc2^{Δ/Y}* brains at all three stages with the largest effect at E18.5 (658 dysregulated genes at FDR < 0.05 and 1299 genes at $p < 0.01$) (Fig. 3a–c and Supplementary Data 3). As there were only 15 significantly dysregulated genes at E14.5 (FDR < 0.05), we opted to use a cut off criteria of $p < 0.01$ at all-time points for rank-based gene ontology enrichment analysis using g:Profiler[39]. The differentially expressed genes (with $p < 0.01$ cut off) in *Thoc2^{Δ/Y}* E14.5 brains were enriched in processes like transcription, histone methylation and embryonic development among others (Fig. 3d and Supplementary Fig. 3b). However, GO analysis of E18.5 dysregulated transcriptome showed enrichment of genes in processes of apoptosis, cell cycle regulation, response to DNA damage, learning and memory as well as RNA Polymerase II associated transcription (Fig. 3e and Supplementary Fig. 3c). Finally, the dysregulated genes in *Thoc2^{Δ/Y}* P10 brains showed GO term enrichment for RNA Polymerase II associated transcription, mitosis, and cell death; the terms also shared with the E14.5 and E18.5 stages (Fig. 3f and Supplementary Fig. 3d). Interestingly, however, the dysregulated genes from only E18.5 and P10 showed an enrichment for genes with higher GC content compared to all expressed genes in the mouse genome (Fig. 3g).

We found that the dysregulated genes were significantly over represented in a list of 1571 known NDD genes (SysNDD)[40] at all developmental stages; 29/225 in E14.5 ($p < 0.001$), 185/1299 in E18.5 ($p < 1.9e{-}20$) and 56/618 in P10 ($p < 0.04$) of *Thoc2^{Δ/Y}* brains (the $p$ values represent hypergeometric test significance) (Supplementary Fig. 3e and Supplementary Data 3). We utilized mammalian phenotype ontology developed by Mouse Genome Informatics (MGI) and Knockout Mouse Phenotyping Program 2 (KOMP2) to further analyse the ontological enrichment of significantly dysregulated genes in E14.5, E18.5 and P10 *Thoc2^{Δ/Y}* mice brains. The dysregulated genes in E14.5 were enriched for genes associated with phenotypes such as completely penetrant pre-weaning lethality, abnormal lung and eye morphology, abnormal embryo size, decreased bone mass and body length and reduced fertility (Supplementary Fig. 3f). Similarly, E18.5 dysregulated genes showed enrichment for genes associated with phenotypes such as embryonic growth retardation and lethality during foetal growth, pre-weaning lethality, spermatogenesis defect, and decreased body weight among others (Supplementary Fig. 3g). Finally, the P10 dysregulated genes were enriched for processes like myelin sheath morphology, oligodendrocyte physiology, and abnormal Schwann cell morphology along with phenotypes such as decreased body weight, embryonic growth retardation and incompletely penetrant pre-weaning lethality (Supplementary Fig. 3h).

Together, our developmental transcriptome data highlighted that the THOC2 exon 37-38 truncated protein has a major impact on transcription, cell death, and cell cycle in *Thoc2^{Δ/Y}* mouse brains at all three developmental stages, with the E18.5 stage being the most affected. The phenotypic ontology terms, embryonic lethality, reduction in body weight along with learning and memory impairment are consistent with our IVF and neurobehavioral testing data.

### *Thoc2^{Δ/Y}* E18.5 and P10 mouse brains show significant dysregulation of proteins involved in mRNA processing, synapse organization and cognitive processes

With significant transcriptional dysregulation in the *Thoc2^{Δ/Y}* E14.5, E18.5 and P10 brains, we set out to explore the impact of truncated THOC2 protein on the *Thoc2^{Δ/Y}* brain proteome using quantitative liquid chromatography coupled to tandem mass spectrometry analysis (LC-MS/MS). We observed significant protein alterations in both E18.5 (644 proteins) and P10 (396 proteins) *Thoc2^{Δ/Y}* mice brains with larger proportion of proteins down- (E18.5: 75% and P10: 67%) than up-regulated (E18.5: 25% and P10: 33%) (Fig. 4a, b and Supplementary Data 4). Gene ontology analysis showed that significantly dysregulated proteins at E18.5 were enriched for processes such as gene expression regulation, splicing and synapse organization among others (Fig. 4c, d and Supplementary Fig. 4a, b). At P10 stage, the dysregulated proteins were enriched for cellular functions involving spliceosome, mRNA surveillance, pre- and post-synapse organization, and cell junction organization (Fig. 4e, f and Supplementary Fig. 4c, d). Further ontological analysis of proteins significantly dysregulated in both the E18.5 and P10 *Thoc2^{Δ/Y}* mouse brains (using MGI mammalian phenotype and KOMP2 mouse phenotype as the reference databases) revealed an enrichment for processes involved in pre-weaning and embryonic lethality and abnormal behaviours including hyperactivity and spatial learning and memory among others (Fig. 4g–j). Strikingly, 112 dysregulated proteins in E18.5 and 63 in P10 brains were coded by the known NDD genes (Supplementary Fig. 4e, f and Supplementary Data 4)[40]. Furthermore, genes encoding the dysregulated proteins in E18.5 and P10 *Thoc2^{Δ/Y}* mouse brain were enriched for higher GC content (Fig. 4k), a finding similar to that from our RNA-seq data (see Fig. 3g).

We also compared the changes in expression of proteins from E18.5 to P10 in the *Thoc2^{Δ/Y}* and *Thoc2^{+/Y}* mouse brains. Whereas the majority (1581, FDR < 0.05) of proteins showed similar trend of change at these two stages, 421 and 688 proteins behaved differently in *Thoc2^{Δ/Y}* and *Thoc2^{+/Y}* mouse brains, respectively (Supplementary Fig. 4g). Gene ontology analysis of the 688 unique *Thoc2^{+/Y}* proteins showed enrichment mainly for biological processes like protein dephosphorylation, inner mitochondrial membrane organization, cytokinesis, chemical synaptic transmission, and cytosolic transport (Supplementary Fig. 4h). However, 421 unique *Thoc2^{Δ/Y}* proteins showed enrichment for processes like translation, peptide biosynthesis and protein metabolism, gene expression and mRNA catabolic processes (Supplementary Fig. 4i), suggesting that proteins altered in the *Thoc2^{Δ/Y}* are associated with processes different to those in the *Thoc2^{+/Y}* mouse brain. Taken together, both embryonic and postnatal stages of *Thoc2^{Δ/Y}* mouse brains showed significant dysregulation of proteins enriched for pathways consistent with our neurobehavioral and transcriptome data.

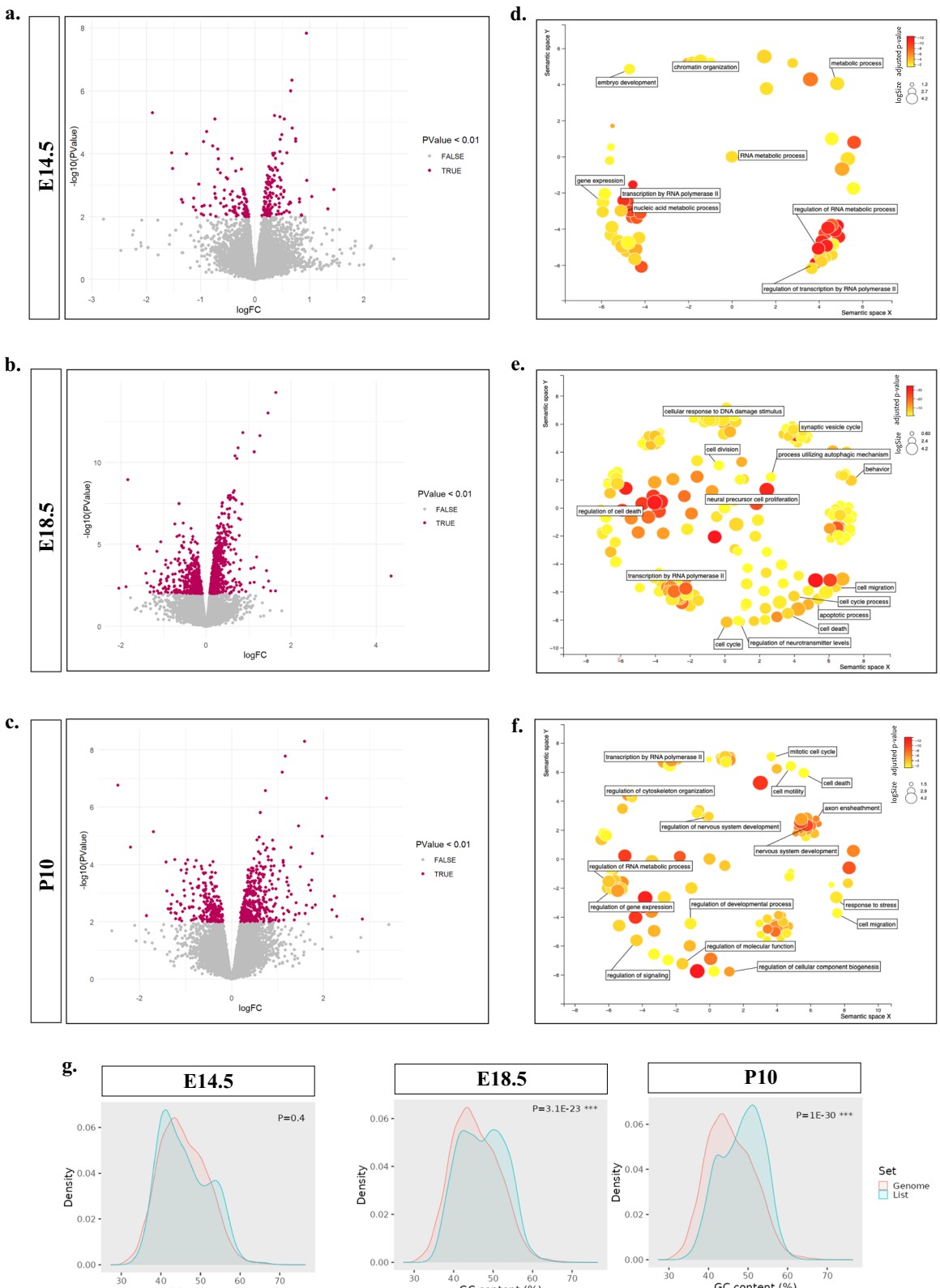

**Fig. 3 | *Thoc2^{Δ/Y}* embryonic and adult mice brains show significantly dysregulated transcriptome.** Volcano plots showing up and downregulated genes in embryonic (E14.5) (**a**), (E18.5) (**b**) and post-natal day 10 (P10) (**c**) *Thoc2^{Δ/Y}* brains with cut-off *p*-value < 0.01 (Quasi-likelihood F-test, two sided). GO BP enrichment analyses shown as ReviGO plots for significant DEGs in E14.5 (**d**), E18.5 (**e**), and P10 (**f**) *Thoc2^{Δ/Y}* mice brains (Fisher's one-tailed hypergeometric tests with Bonferroni correction). **g**, ShinyGO graphs showing comparison of percentage GC content between mouse genome and DEGs from E14.5 (left), E18.5 (middle) and P10 (right) *Thoc2^{Δ/Y}* mice brains (one-sided hypergeometric test for false discovery rate). GO gene ontology, BP biological process, DEG differentially expressed genes. Source data are provided as a Source Data file.

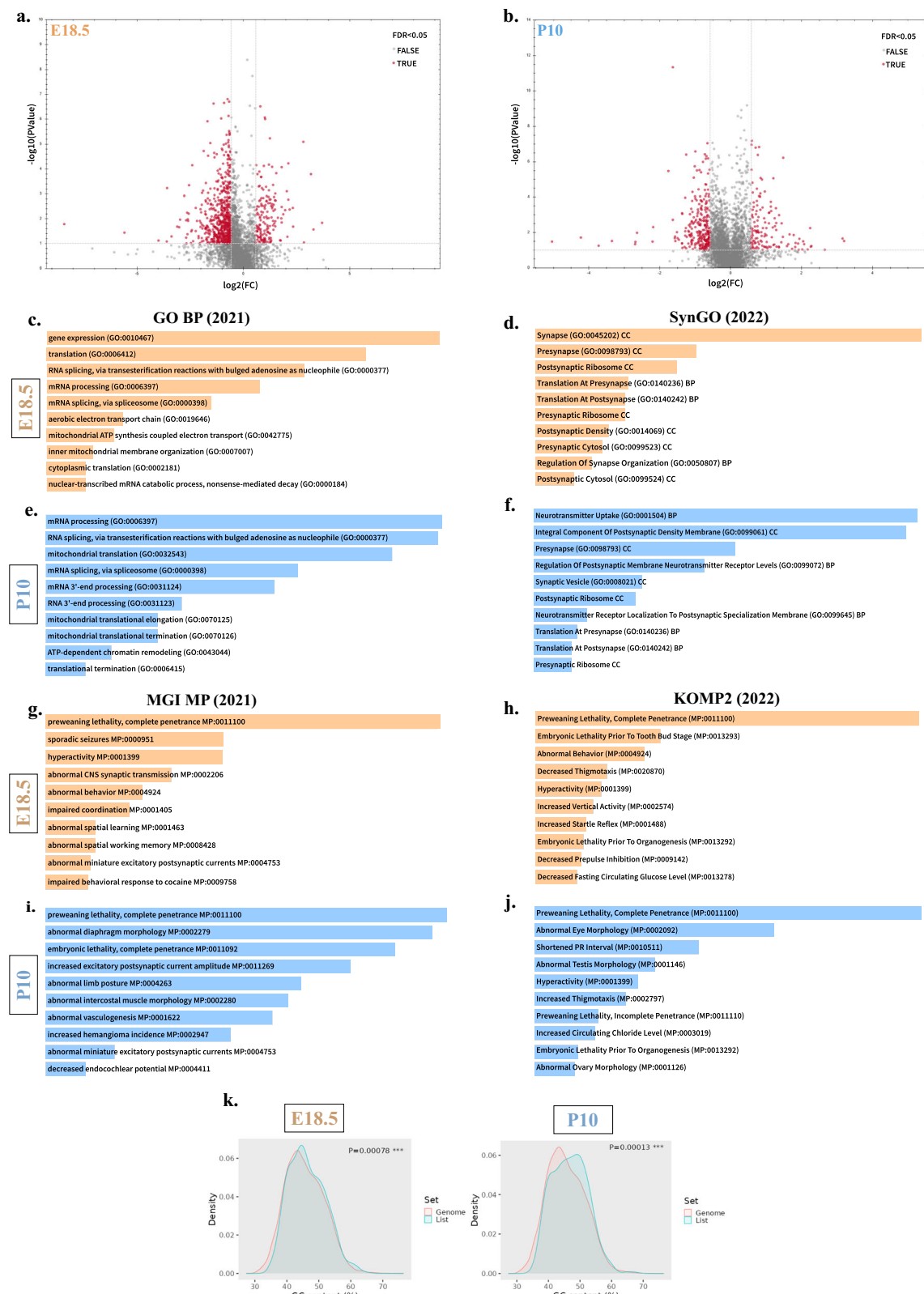

### Thoc2$^{\Delta/Y}$ E18.5 mice embryonic cortex shows reduction in ventricular zone thickness

As the E18.5 stage showed a major transcriptomic dysregulation and is a critical developmental stage representing the completion of neurogenesis in cortical development[41] we performed histological studies on the Thoc2$^{+/Y}$ and Thoc2$^{\Delta/Y}$ E18.5 embryonic brains. While haematoxylin and eosin (H&E) staining did not reveal any gross histological

alterations (Supplementary Fig. 5a, b), a closer inspection of the cortical layers showed a significant reduction in the thickness of the ventricular zone (VZ) of the E18.5 Thoc2$^{\Delta/Y}$ mouse brain (Fig. 5a). We validated these observations by VZ NSC marker PAX6 immunofluorescence staining of E14.5 and E18.5 Thoc2$^{+/Y}$ and Thoc2$^{\Delta/Y}$ mouse brains. We observed a significantly reduced PAX6 layer thickness in both the E14.5 and E18.5 Thoc2$^{\Delta/Y}$ mouse brain (Fig. 5b, c), confirming a

**Fig. 4 | *Thoc2^{Δ/Y}* embryonic and adult mice brains show significantly dysregulated proteome.** Volcano plots showing up- and downregulated proteins in **a**, embryonic (E18.5) and **b**, post-natal day 10 (P10) *Thoc2^{Δ/Y}* mice brains (Unpaired Student's *t* test (two-tailed) for testing differential abundance). The up- and down-regulated proteins have FC > 1.5, and FDR of <0.05 cutoff. GO enrichment analyses of significantly dysregulated proteins in E18.5 (**c**, **d**) and P10 (**e**, **f**) *Thoc2^{Δ/Y}* mice brains performed using Enrichr web-tool. Ontological enrichment analysis using MGI and KOMP2 mouse phenotype databases for significantly dysregulated proteins in E18.5 (**g**, **h**) and P10 (**i**, **j**) *Thoc2^{Δ/Y}* mice brains using Enrichr web-tool. For

**c**–**j** The top 10 enriched GO terms are shown. The length of the horizontal axis represents the enrichment *p* value for the functional clustering. **k** ShinyGO graphs showing comparison of percentage GC content between mouse genome and genes associated with the significantly dysregulated proteins in E18.5 and P10 *Thoc2^{Δ/Y}* mice brains (One-sided hypergeometric test for false discovery rate). FC Fold Change, FDR False Discovery Rate, GO Gene Ontology, BP Biological Process, SynGO Synaptic Gene Ontology, MGI Mouse Genome Informatics, MP Mammalian Phenotype, KOMP2 Knockout Mouse Phenotyping Program 2. Source data are provided as a Source Data file.

reduction in the VZ size. We also performed H&E staining of *Thoc2^{+/Y}* and *Thoc2^{Δ/Y}* mouse brains at P30 and did not find any gross morphology alterations in the *Thoc2^{Δ/Y}* mouse brains compared to *Thoc2^{+/Y}*. However, a closer look at the cytoarchitecture of the cortices showed a significantly reduced overall thickness of the cortical plate and corpus callosum (Fig. 5d, e). Finally, we observed no change in the hippocampus structure or volume (Supplementary Fig. 5c, d). Overall, we found minor, but significant anatomical differences in the VZ of the E18.5 *Thoc2^{Δ/Y}* mouse brains suggesting compromised function of the NSCs.

### *Thoc2^{Δ/Y}* primary NSCs have increased apoptosis, perturbed cell cycle properties and DNA damage

The VZ is the primary proliferative zone that harbors the NSCs[42]. NSC malfunction is frequently involved in NDDs such as primary microcephaly, autism spectrum disorders (ASD) or X-linked ID[43,44]. Reduced VZ thickness and RNA-seq data showing transcriptionally dysregulated genes enriching for cell cycle and apoptotic pathways in E18.5 *Thoc2^{Δ/Y}* brains prompted us to investigate if the NSCs in these mice have altered cell death and/or cell cycle properties. To this end, we analysed the cell cycle kinetics of the primary NSCs from E18.5 *Thoc2^{+/Y}* and *Thoc2^{Δ/Y}* mouse brains using Hoechst staining. We observed a significant increase of *Thoc2^{Δ/Y}* NSCs in the G2/M phase with a concomitant reduction in the G1 phase (Fig. 6a), indicating a delay in the completion of cell cycle in the *Thoc2^{Δ/Y}* NSCs. TUNEL assay showed a significantly higher percentage of TUNEL positive cells indicating increased apoptosis of *Thoc2^{Δ/Y}* compared to *Thoc2^{+/Y}* NSCs (Fig. 6b). Consistent with the TUNEL results, our Annexin-V assay showed a significant increase in apoptotic cells in the *Thoc2^{Δ/Y}* compared to *Thoc2^{+/Y}* NSCs (Fig. 6c).

The G2/M checkpoint control senses unrepaired DNA damage and allows cells to repair before entering mitosis[45]. We asked if the increase in *Thoc2^{Δ/Y}* NSCs in the G2/M phase could be a consequence of checkpoint invoked by accumulation of DNA damage in the preceding phase/s and could represent a pool of NSCs that, for unknown reasons, did not undergo apoptosis. We checked this possibility by determining the levels of classical DNA damage marker i.e., γ-H2AX protein (H2AX phosphorylated at Ser139)[46] in *Thoc2^{+/Y}* and *Thoc2^{Δ/Y}* NSCs by western blotting. We found significantly higher γ-H2AX levels in *Thoc2^{Δ/}* than the *Thoc2^{+/Y}* NSCs although the total H2AX protein levels were unchanged (Fig. 6d). We validated this observation by analyzing the amount of DNA damage in NSCs using single cell gel electrophoresis comet assay[47]. Consistent with increased γ-H2AX levels, a significantly higher number of *Thoc2^{Δ/Y}* NSCs showed comet tail formation compared to the *Thoc2^{+/Y}* NSCs, indicating accumulation of fragmented DNA resulting from DNA damage (Fig. 6e and Supplementary Fig. 6a, b). We next asked if the DNA damage is also present in vivo. To address this, we determined γ-H2AX and H2AX protein levels in *Thoc2^{Δ/Y}* and *Thoc2^{+/Y}* E14.5 and E18.5 (the stages that contain high number of NSCs) embryonic brain lysates. Similar to the in vitro NSCs, E14.5 and E18.5 *Thoc2^{Δ/Y}* mouse brains showed significantly increased γ-H2AX but not the H2AX protein levels (Fig. 6f). Lastly, we performed comet assay on the *THOC2* exon 37-38 deletion patient fibroblasts to determine if these cells also have damaged DNA. Notably, compared to the unrelated

human control fibroblasts, we observed comet tail formation in a significant number of patient fibroblasts, indicating an elevated DNA damage in patient cells (Fig. 6g and Supplementary Fig. 6c, d). Together, these data showed that *THOC2* exon 37-38 deletion causes DNA damage both in the patient fibroblasts and the *Thoc2^{Δ/Y}* mouse NSCs.

### *Thoc2^{Δ/Y}* NSCs and *THOC2* exon 37-38 deletion patient fibroblasts have increased R-loops and DNA Damage

R-loops are three-stranded nucleic acid structures, composed of DNA:RNA hybrids and the associated non-template single-stranded DNAs that are highly susceptible to DNA breaks[48,49]. To determine if the DNA damage in the *Thoc2^{Δ/Y}* NSCs was due to R-loop accumulation, we probed *Thoc2^{+/Y}* and *Thoc2^{Δ/Y}* NSCs with anti-RNA-DNA hybrid S9.6 monoclonal antibody to detect formation of R-loops by immunofluorescence. We observed a significantly increased accumulation of RNase H-sensitive R-loops with characteristic S9.6 fluorescent nuclear foci in *Thoc2^{Δ/Y}* compared to *Thoc2^{+/Y}* NSCs as well as patient compared to control fibroblasts (Fig. 7a, b). We observed similar results, albeit not as typical nuclear R-loop foci, using a different S9.6 antibody (Supplementary Fig. 7a, b). As RNase T1 treatment specifically degrades single-stranded RNAs, leaving RNA:DNA hybrids intact, we detected a clear increase in S9.6 nuclear signal in *Thoc2^{Δ/Y}* compared to *Thoc2^{+/Y}* RNase T1 treated NSCs as well as in patient compared to control fibroblasts, suggesting accumulation of RNA:DNA hybrids (Supplementary Fig. 7a, b)[50,51]. The RNase H treatment, that specifically digests RNA in RNA:DNA hybrids, abolished S9.6 signal both in *Thoc2^{Δ/Y}* and *Thoc2^{+/Y}* NSCs and in control and patient fibroblasts (Fig. 7a, b and Supplementary Fig. 7a, b). We further confirmed these data by S9.6 dot-blot assay on the genomic DNA extracted (in RNase-free conditions) from mouse NSCs and patient fibroblasts (see Supplementary Fig. 7c for loading). While we observed no difference in S9.6 signal between the untreated *Thoc2^{Δ/Y}* and *Thoc2^{+/Y}* NSCs (Fig. 7c, top panel), RNase T1 treatment resulted in overall reduced but higher S9.6 signal in the *Thoc2^{Δ/Y}* than *Thoc2^{+/Y}* NSCs implying an accumulation of the R-loops in the *Thoc2^{Δ/Y}* NSCs (Fig. 7c, middle panel). As expected, S9.6-specific signal was abolished in the RNase H treated *Thoc2^{Δ/Y}* and *Thoc2^{+/Y}* NSCs (Fig. 7c, bottom panel). The untreated, RNase T1 or RNase H treated *THOC2* exon 37-38 deletion patient and control fibroblasts showed S9.6 signals similar to those observed in the *Thoc2^{Δ/Y}* and *Thoc2^{+/Y}* NSCs, respectively, indicating R-loops also accumulate in the patient cells (Fig. 7d). We premised if reducing R-loop accumulation would be sufficient to rescue the DNA damage phenotype of *Thoc2^{Δ/Y}* NSCs. Indeed, RNase H1 overexpression resulted in significantly reduced R-loop accumulation (Fig. 7e) and comet tail formation (Fig. 7f) in *Thoc2^{Δ/Y}* NSCs, indicating that R-loop accumulation is likely responsible for DNA damage in these cells. Taken together, we suggest that *THOC2* exon 37-38 deletion interferes with R-loop homeostasis and contributes to the increased DNA damage in the *Thoc2^{Δ/Y}* NSCs and patient fibroblasts.

### *Thoc2^{Δ/Y}* neurons show perturbed in vitro neural migration, electrophysiological properties and synapse formation

The molecular and cellular alterations in *Thoc2^{Δ/Y}* NSCs, together with the transcriptomic and proteomic data indicating multiple

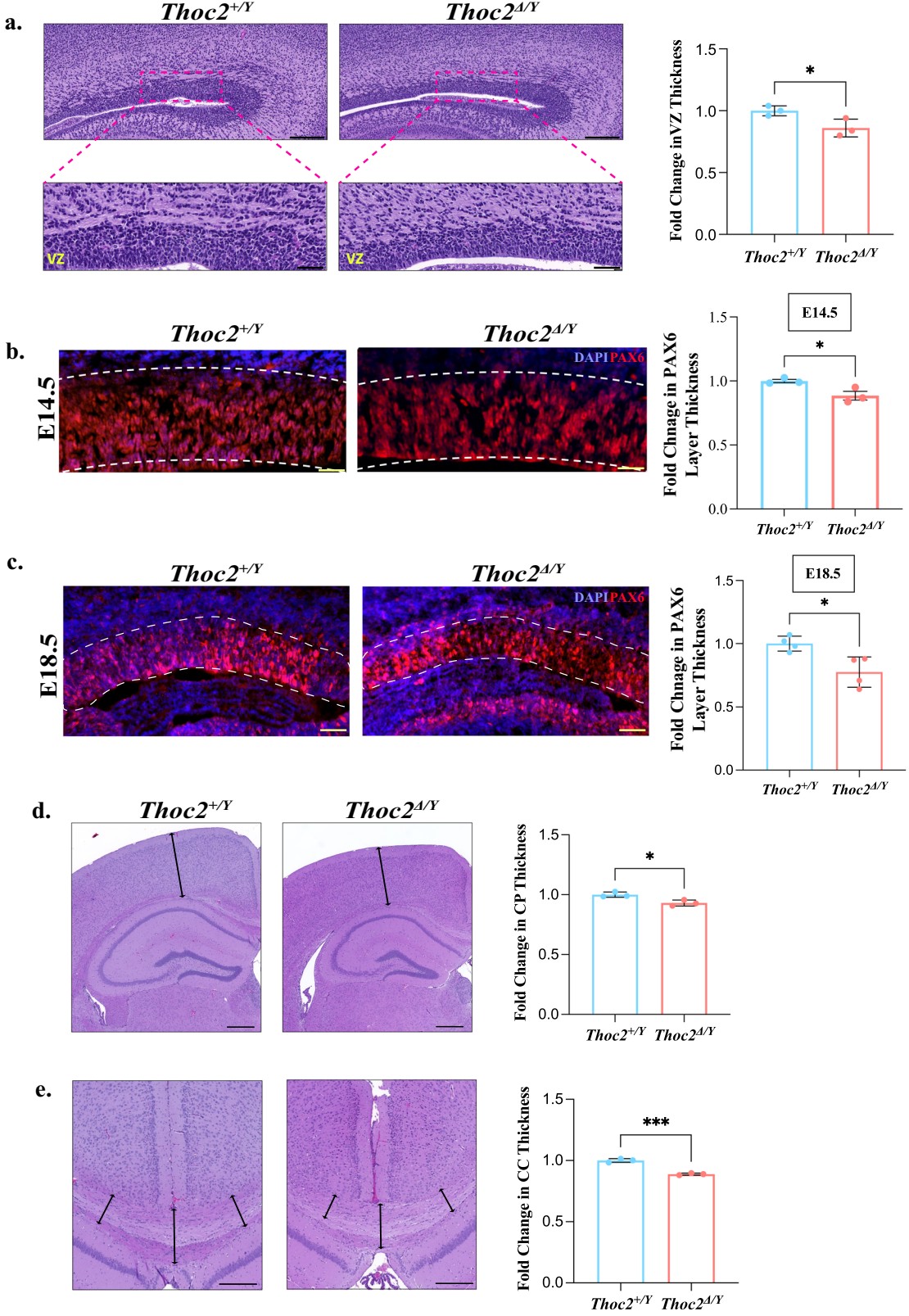

perturbations in neuronal processes, prompted us to investigate the effect of *Thoc2* exon 37-38 deletion on neuron structure and function. We hypothesized that neurons, developing from the defective NSCs, might also show DNA damage. We addressed this by γ-H2AX immunofluorescence staining of the cortical/hippocampal *Thoc2*[Δ/Y] E18.5 mouse primary neurons. We found that significant number of *Thoc2*[Δ/Y] neurons show increased γ-H2AX foci, indicating increased DNA

damage (Fig. 8a). As NSC perturbations can lead to altered neuronal migration[52,53], we asked if *Thoc2*[Δ/Y] also impacts neuronal migration. We tested this by an in vitro neuronal migration assay using *Thoc2*[+/Y] and *Thoc2*[Δ/Y] neurospheres[52,54]. We observed a significant reduction in distance the neurons migrated out of the seeded neurospheres from *Thoc2*[Δ/Y] compared to those from *Thoc2*[+/Y] mouse brain (Fig. 8b and Supplementary Fig. 8a). As the evidence suggests that different cell

**Fig. 5 | Altered cytoarchitecture in *Thoc2*$^{\Delta/Y}$ mouse brain. a** H&E staining of E18.5 embryonic brain coronal sections. The boxed VZ area is shown in a magnified view in the lower panel. VZ: Ventricular Zone. Scale bar: upper panels, 200 μm; lower panels, 50 μm. Graph showing quantitative measurement of VZ thickness. *Thoc2*$^{+/Y}$ to *Thoc2*$^{\Delta/Y}$ fold change is plotted from mean values ± SEM ($n = 3$ embryos per genotype); *$p = 0.04$. Immunofluorescent staining for PAX6 in E14.5 ($n = 3$) (**b**) and E18.5 ($n = 4$) (**c**) embryonic brain coronal sections. VZ is marked with white dotted lines. Scale Bar: **b** 50 μm; **c** 200 μm. Graph showing quantitative measurement of VZ region as PAX6$^+$ layer thickness. *Thoc2*$^{+/Y}$ to *Thoc2*$^{\Delta/Y}$ fold change plotted from mean values ± SEM; *$p = 0.03$ (**b**) and 0.02 (**c**). **d** Representative images of H&E staining of adult (30 days) mouse brain coronal sections used for cortical plate (CP) thickness measurement. Scale Bar: 250 μm. Graphs showing quantitative measurement of CP thickness. *Thoc2*$^{+/Y}$ to *Thoc2*$^{\Delta/Y}$ fold change plotted using mean values ± SEM ($n = 3$ mice per genotype); *$p = 0.022$. **e** Representative images of H&E staining of adult (30 days) mouse brain coronal sections used for measurement of corpus callosum (CC) thickness at hippocampal level. Scale Bar: 100 μm. Graph showing quantitative measurement of CC thickness. *Thoc2*$^{+/Y}$ to *Thoc2*$^{\Delta/Y}$ fold change plotted using mean values ± SEM ($n = 3$ mice per genotype); ***$p = 0.0003$. Two-tailed unpaired student's $t$ tests were used in all figures. Source data are provided as a Source Data file.

fates might be determined by the dynamics of the preceding rounds of NSC divisions[55], we asked if the altered cell cycle properties in *Thoc2*$^{\Delta/Y}$ NSCs impacts neural differentiation. To address this, we performed in vitro neural differentiation assay on E18.5 *Thoc2*$^{\Delta/Y}$ and *Thoc2*$^{+/Y}$ NSCs[56]. We then stained the cells in culture with cell specific markers (NSC: PAX6; Neurons: ß-III TUBULIN; Astrocytes: GFAP and Oligodendrocytes: CNPASE) (Fig. 8c) and scored the different cell types. We observed a significant increase in the differentiated cell population in the *Thoc2*$^{\Delta/Y}$ as compared to *Thoc2*$^{+/Y}$ cultures, indicating a premature differentiation of the *Thoc2*$^{\Delta/Y}$ NSCs. Consistent with this observation, we observed a significant concomitant reduction in the number of progenitor cells in the *Thoc2*$^{\Delta/Y}$ as compared to *Thoc2*$^{+/Y}$ cultures (Fig. 8c). We also analyzed the primary axon length, an indicator of the connectivity potential of neurons, of *Thoc2*$^{+/Y}$ and *Thoc2*$^{\Delta/Y}$ E18.5 mouse neurons. While the *Thoc2*$^{\Delta/Y}$ neurons had 32% shorter primary axons as compared to the *Thoc2*$^{+/Y}$ neurons (Fig. 8d), no significant changes were observed in the number of neurite termini between the *Thoc2*$^{+/Y}$ and *Thoc2*$^{\Delta/Y}$ neurons (Supplementary Fig. 8b).

As we observed significant learning, memory, and sensorimotor problems as well as perturbations in neural maturation in the *Thoc2*$^{\Delta/Y}$ mice, we asked if these outcomes could be due to changes in synapse formation and/or function. We determined this by immunofluorescent staining for the pre- and post-synaptic markers Synapsin-1 (SYN1) and postsynaptic density-95 (PSD95) proteins, respectively, followed by colocalization quantification of excitatory synaptic puncta[57]. We observed a significant reduction in SYN1-PSD-95 puncta in *Thoc2*$^{\Delta/Y}$ compared to the *Thoc2*$^{+/Y}$ days in vitro 14 (DIV14) neurons, indicating an impairment in formation of functional synaptic connections among the *Thoc2*$^{\Delta/Y}$ DIV14 neurons (Fig. 8e). We also analysed in vivo synapse forming potential of *Thoc2*$^{\Delta/Y}$ mouse neurons using Golgi-Cox staining of the P30 *Thoc2*$^{\Delta/Y}$ mice brains and analysing the dendritic spines. While there was no obvious change in dendritic spine density between *Thoc2*$^{\Delta/Y}$ and *Thoc2*$^{+/Y}$ neurons (Supplementary Fig. 8c), we observed a significant reduction in mature spines in the *Thoc2*$^{\Delta/Y}$ neurons (Fig. 8f) consistent with our in vitro synaptic puncta results Fig. 8e.

The morphological and synaptic impairments prompted us to investigate if there were any functional electrophysiological alterations in the *Thoc2*$^{\Delta/Y}$ mouse neurons. We tested this possibility using multi-electrode arrays (MEA) by determining the electrical activity of DIV7 and DIV14 primary cortical/hippocampal neuron cultures generated from E18.5 *Thoc2*$^{+/Y}$ and *Thoc2*$^{\Delta/Y}$ mouse brains. DIV7 neurons, that are at the immature stage in culture, did not show any difference in electrophysiological parameters like mean firing rate, mean burst rate, mean burst spike rate and network burst rate between the *Thoc2*$^{\Delta/Y}$ and *Thoc2*$^{+/Y}$ neuron cultures (Supplementary Fig. 8d). However, at DIV14, a more mature stage of neuronal development, we observed significant reduction in neural network activity as indicated by little to no network bursts in the *Thoc2*$^{\Delta/Y}$ neuronal networks (Fig. 8g and Supplementary Fig. 8e) in neurons plated at same density (Supplementary Fig. 8f). This observation was confirmed by an increase in random spike (isolated, a-synchronous spikes) frequency in *Thoc2*$^{\Delta/Y}$ cultures, indicative of failure to form mature neuronal network activity (Fig. 8h). Apart from the neural network activity, the overall electrical activity of the neurons cultured from *Thoc2*$^{\Delta/Y}$ mice brains was also significantly decreased, as indicated by the reduction in mean firing rate, mean burst rate and mean burst spike rate (Fig. 8i–l).

Overall, we conclude that *Thoc2* exon 37-38 deletion causes multiple downstream impairments of NSC-derived neurons like migration, maturation, synapse formation and electrophysiology. Our observation of significant reduction in excitatory synaptic puncta formation is consistent with the data showing failure of *Thoc2*$^{\Delta/Y}$ E18.5 mouse neurons in establishing synchronous functional neural networks. These results were supported by our transcriptome and proteome data showing enrichment of dysregulated gene/proteins in synapse formation and function.

## Discussion

Originally identified as mRNA export machinery, the TREX complex has emerged to have roles in multiple biological processes including transcription regulation, maintaining genomic stability, and regulating stem cell maintenance[6,7,16,58]. Accumulating evidence has linked perturbed TREX function to neurodevelopmental and neurodegenerative disorders as well as cancer[5,7,59–62]. As an essential TREX subunit, THOC2 has been the focus of many studies[11,59,60,63–67]. While these studies have provided insights into varied roles of THOC2 protein, the findings were obtained mainly from in vitro knockdown cell or knockout lower organism models[6,20–22,24,25,68]. The lack of a viable mammalian model has been a major barrier to investigating the biological significance of THOC2 (and more broadly the TREX complex) due to gene essentiality. Thus, to gain deeper insights into the biology of THOC2 and TREX in brain development, and more specifically, how compromised *THOC2* function leads to NDDs, we generated and studied the mouse model of *THOC2* syndrome based on a pathogenic exon 37-38 deletion patient variant[5].

The yeast Tho2 C-terminal region has been shown to bind nucleic acids and facilitate recruitment of the THO to chromatin[10]. While Tho2 deleted for the C-terminal region did not affect TREX assembly, it did impact gene expression[10]. The *Thoc2* exon 37-38 deletion also did not have a significant effect on the levels of THO/TREX subunits (see Supplementary Fig. 1e), but did impact gene expression and genome stability, suggesting a partial loss of function mechanism of THOC2 exon 37-38 deleted protein in the mouse brain. Overall, the *Thoc2*$^{\Delta/Y}$ mice phenocopied the core features of the *THOC2* syndrome. Similar to the patient, the *Thoc2*$^{\Delta/Y}$ mice had reduced body weight and size. In addition, the birth rate of *Thoc2*$^{\Delta/Y}$ mice was reduced by about one third of the number expected. Therefore, we investigated if reduction in body size and weight across all organs as well as low birth rate was due to general and early developmental deficits in proliferation, differentiation, and likely cell death of stem cells.

IVF assays showed that the lower birth rate of *Thoc2*$^{\Delta/Y}$ mice is likely due to alterations before the blastocyst stage, when the majority of the cells in the embryo are multipotent stem cells. A previous report has also shown *Thoc2* to be important for blastocyst development and pluripotency maintenance[6]. Together, these findings confirmed the crucial role of *Thoc2* during early embryonic development, and likely in maintaining the stem cell niche. That THO subunit proteins are also

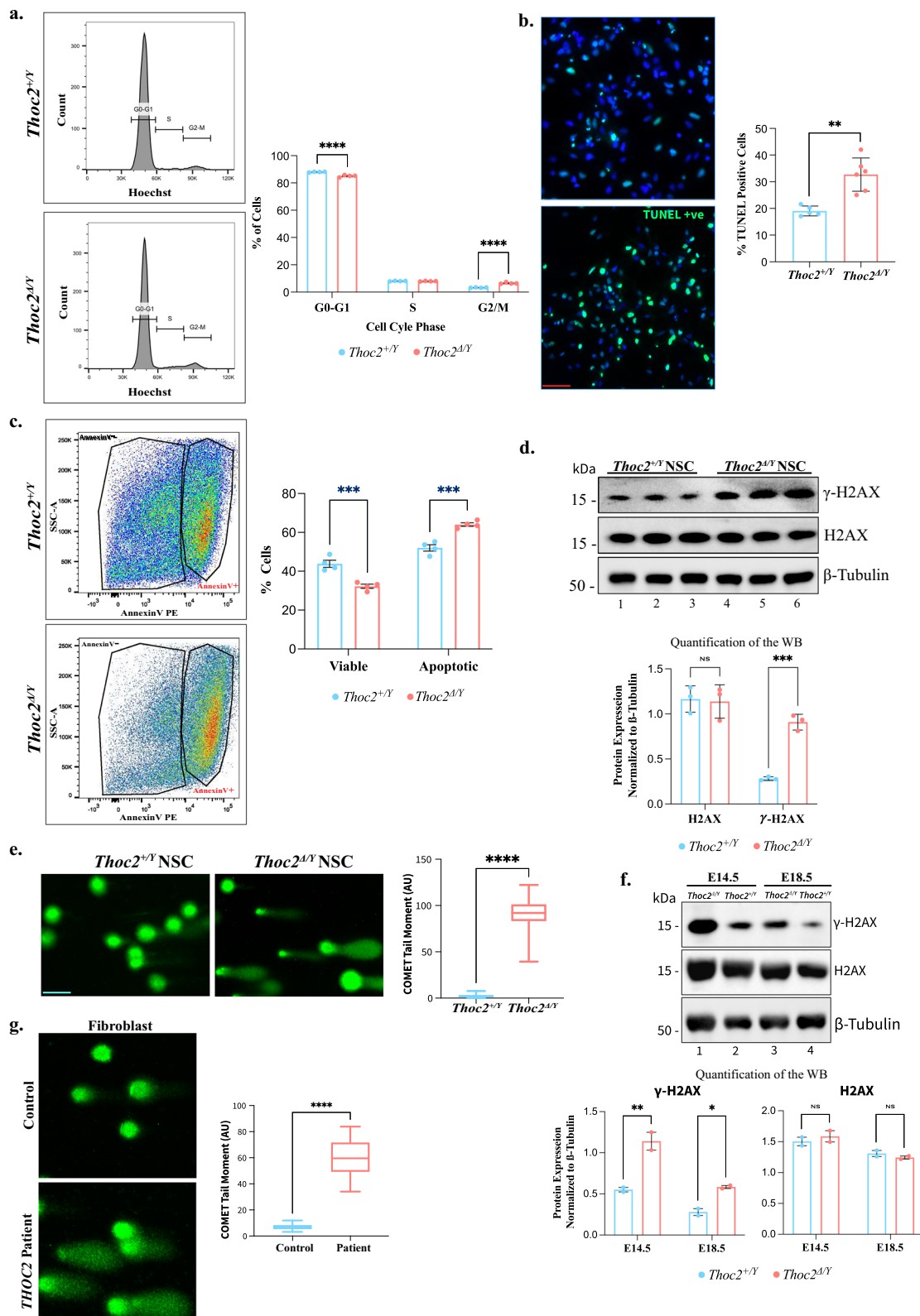

essential for maintenance of haematopoiesis, intestinal stem cells and triple negative breast cancer stemness indicates broader developmental roles of the TREX complex[59,69,70]. The importance of the THO complex in brain development was supported by our observations that THOC2 and virtually all other TREX subunit protein levels are highly abundant in brains at embryonic stages (E14.5 and E18.5) but decline sharply from post-natal day 10 (P10) to P60. (see Fig. 1 and

Supplementary Fig. 1). The precise molecular events that allow some *Thoc2^(Δ/Y)* mouse embryos to survive to birth and beyond are unknown and need further investigations.

The *Thoc2^(Δ/Y)* mice that survived, however, generally recapitulated the phenotype of *THOC2* syndrome[3–5]. Multiple mouse models of ID and ASDs for genes like *Syngap2*, *Fmr1*, *Mecp2* and *Shank2/3* with converging cognitive, social, and emotional deficits have proved to be

**Fig. 6 | Perturbed cell cycle kinetics, increased cell death and elevated levels of DNA damage in $Thoc2^{\Delta/Y}$ mice. a** Representative flow cytometry plots showing gating for Hoechst-stained NSCs at different stages of cell cycle. Graph showing percentage of cells at different stages of cell cycle; $n = 5$ embryos per genotype and 3 independent experiments; ****$p < 0.0001$; two-way ANOVA, Bonferroni's multiple comparison test. **b** Representative images of TUNEL-stained nuclei (cells stained green). Scale Bar 100 μm. Quantification of TUNEL positive cells between $Thoc2^{\Delta/Y}$ ($n = 6$) and $Thoc2^{+/Y}$ ($n = 5$) NSCs: **$p = 0.0011$; two-tailed unpaired student's $t$ test. **c** Representative scatter plots showing gating for Annexin V positive and negative NSCs. Graph showing percentage of viable (Annexin V−) and apoptotic (Annexin V+) cells; $n = 4$ embryos per genotype and 3 independent experiments; ***$p = 0.0001$; two-way ANOVA, Bonferroni's multiple comparison test. **d** Representative western blot showing γ-H2AX and H2AX proteins in $Thoc2^{\Delta/Y}$ and $Thoc2^{+/Y}$ NSCs. β-Tubulin was used as loading control. Graph showing γ-H2AX and H2AX levels relative to housekeeping β-Tubulin protein quantified from the western blots; $n = 3$ embryos per genotype; NS non-significant; ***$p = 0.0006$; two-way ANOVA, Bonferroni's multiple comparison test. **e**, Representative images of alkaline comet assay of $Thoc2^{+/Y}$ and $Thoc2^{\Delta/Y}$ NSCs. Box plot showing comet tail moment quantified from $Thoc2^{+/Y}$ and $Thoc2^{\Delta/Y}$ NSCs (Scale Bar 50 μm); $n = 4$ embryos per genotype; ****$p = 0.000003$; two-tailed unpaired student's $t$ test. **f** Western blot showing γ-H2AX and H2AX proteins in $Thoc2^{+/Y}$ and $Thoc2^{\Delta/Y}$ embryonic brain (E14.5 and E18.5). β-Tubulin was used as loading control. Graph showing γ-H2AX and H2AX levels relative to housekeeping β-Tubulin protein from western blots; $n = 2$ embryos and 3 independent experiments); **$p = 0.004$; *$P = 0.046$; two-way ANOVA, Bonferroni's multiple comparison test. **g**, Representative images of alkaline comet assay on control fibroblast and $THOC2$ exon 37-38 deletion patient fibroblasts (Scale Bar 50 μm). Box plot showing comet tail moment quantified from control and $THOC2$ exon 37-38 deletion patient fibroblasts; $n = 3$ independent experiments; ****$p < 0.0001$; two-tailed unpaired student's $t$ test. All the data presented are mean values ± SEM. Source data are provided as a Source Data file.

excellent tools for understanding the neuropathology (reviewed in[71]). Likewise, the $Thoc2^{\Delta/Y}$ mice showed significant deficit in learning the task on both the Morris Water Maze and Barnes maze, as well as impaired spatial working and reference memory on the Y-maze. Such cognitive impairments are clinical hallmarks for NDD/ID disease models[71], and suggest that the $Thoc2^{\Delta/Y}$ mouse model has translational relevance for $THOC2$ syndrome, where patients have ID as the core clinical presentation[5]. Interestingly, the $Thoc2^{\Delta/Y}$ mice are also hyperactive and have decreased anxiety-associated behaviour, traits seen as comorbid features in a subset of $THOC2$ patients[5]. These observations are consistent with our proteomics data, which showed dysregulation of proteins associated with these behavioural phenotypes (see Fig. 4 and Supplementary Data 4). Moreover, we observed significant deficits in sensorimotor control and fine motor coordination noted on multiple behavioural measures, including the pasta handling task, the adhesive removal test and beam walking in $Thoc2^{\Delta/Y}$ mice. While the neural basis of this has yet to be explored, it may be due to impaired myelination in these mice. In line with this, a significant generalised delay in myelination and reduction in white matter volume, particularly in the trigone region was observed from MRI imaging in the $THOC2$ exon 37-38 deletion patient[5]. Importantly, however, overall gross motor functions in $Thoc2^{\Delta/Y}$ mice were not impaired, as indicated by performance comparable to wild-type mice on the rotarod, Digigait, and grip strength tests. This is similar to what is observed in $THOC2$ patients[5], and is in line with imaging findings in the $THOC2$ exon 37-38 deletion patient, which indicated moderate plagiocephaly and brachycephaly, but no $ex$-$vacuo$ dilation of the lateral ventricles, no heterotopic grey matter and normal-appearing subcortical and brainstem structures. Similarly, the E18.5 $Thoc2^{\Delta/Y}$ mouse brains had reduced VZ but no other gross morphological anomaly. However, our transcriptomic and proteomic data from the embryonic and adult $Thoc2^{\Delta/Y}$ mouse brains revealed significant dysregulation of genes and proteins that normally are major mediators of nervous system function and development. Enrichment of dysregulated genes/proteins in pathways like transcription, splicing, embryonic lethality, learning, memory, and synapse organization converge very well with our cellular and molecular findings, e.g., death of $Thoc2^{\Delta/Y}$ mice before blastocyst stage, the behavioural tests showing significant perturbation in learning and memory and cell cycle anomaly and cell death of the $Thoc2^{\Delta/Y}$ mouse NSCs. Interestingly, we observed significant transcriptional dysregulation of known NDD genes, like $Syngap1$, $Huwe1$, $Shank3$, $Dlg4$, $Kdm5c$, and $Ctnnd1$ (among many others) (see Supplementary Fig. 3), which indicates that $Thoc2$ acts as an important regulator of neurodevelopmental processes and its compromised function results in diverse neurological symptoms.

An increased THOC2 protein abundance during embryonic stages and significant death of $Thoc2^{\Delta/Y}$ embryos before blastocyst stage, together with the data showing that THOC2 and THOC5 are required for mESC maintenance[6], strongly support a critical role of THOC2/

TREX in early development. These findings, together with a reduced VZ thickness in the E14.5 and E18.5 $Thoc2^{\Delta/Y}$ mouse brains and neuropathological outcomes pointed towards likely aberrations in the neural progenitors. We noted deregulation of the cell cycle, particularly at the G2/M phase, in the $Thoc2^{\Delta/Y}$ NSCs. The G2/M checkpoint is activated upon DNA damage and prevents the cells from entering mitosis with damaged genomic DNA[45]. Indeed, $Thoc2^{\Delta/Y}$ mouse NSCs had higher amounts of damaged DNA. The elevated DNA damage was followed by increased apoptosis in the $Thoc2^{\Delta/Y}$ NSCs. Note that a recent study has also reported an increased G2/M cell count in $THOC2$ and $THOC5$ depleted radioresistant triple negative breast cancer (TNBC) cells[59].

Impairment of NSCs has been linked to multiple NDD, neurodegenerative disorders, and to brain tumours [reviewed in ref. 43]. The multiple molecular and cellular impairments observed in $Thoc2^{\Delta/Y}$ NSCs were correlated with a plethora of neuron structure and function anomalies, for example, presence of significant in vitro neural migration defects with premature differentiation of NSCs and axon shortening in the cultured $Thoc2^{\Delta/Y}$ primary neurons. Notably, we found a significant reduction in mature dendritic spines in the $Thoc2^{\Delta/Y}$ mice brains and a reduction in formation of synaptic puncta in cultured $Thoc2^{\Delta/Y}$ mouse primary neurons, which agree with the results showing an impaired neural network activity in the $Thoc2^{\Delta/Y}$ mouse neurons in DIV14 culture. Our results corroborate with synaptic disorganizations observed in NDDs like ID, ASD, and epilepsy[72,73] along with studies showing minimal alteration in proper synaptic activity causing significant disruption in the neuronal network formation[74,75]. Interestingly, our proteomics data also showed significant dysregulation of proteins associated with synapse formation and function. Taken together, our data showing multiple functional deficits in $Thoc2^{\Delta/Y}$ mouse neurons suggest that $THOC2$ has critical roles in post-mitotic neurons. Literature also shows that TREX complex plays an essential role in cellular senescence[76]. Furthermore, THOC2 was shown to mislocalize in Huntington's Disease and Amyotrophic lateral sclerosis, potentially contributing to these neurodegenerative pathologies[26]. Therefore, neuronal defects in our mouse model together with the other studies highlight a critical role for THOC2/TREX in neurons, stem cells and early stages of mouse development.

Studies have shown that increased DNA damage in THO subunit depleted cells was caused by unscheduled accumulation of triple-stranded R-loop structures[19,77], and that other TREX components were also close interactors of R-loops[78]. In addition, with their co-transcriptional origin and increased susceptibility to DNA damage, accumulating recent evidence has recognized R-loops as potential regulators of gene expression[79,80]. More recently, R-loops and associated DNA damage have been implicated in the etiology of neurological disease like spinal muscular atrophy (SMA) and ALS[81]. Based on these results and the evidence that the yeast Tho2 (orthologue of mammalian $THOC2$) interacts with nucleic acids via it's C-terminal

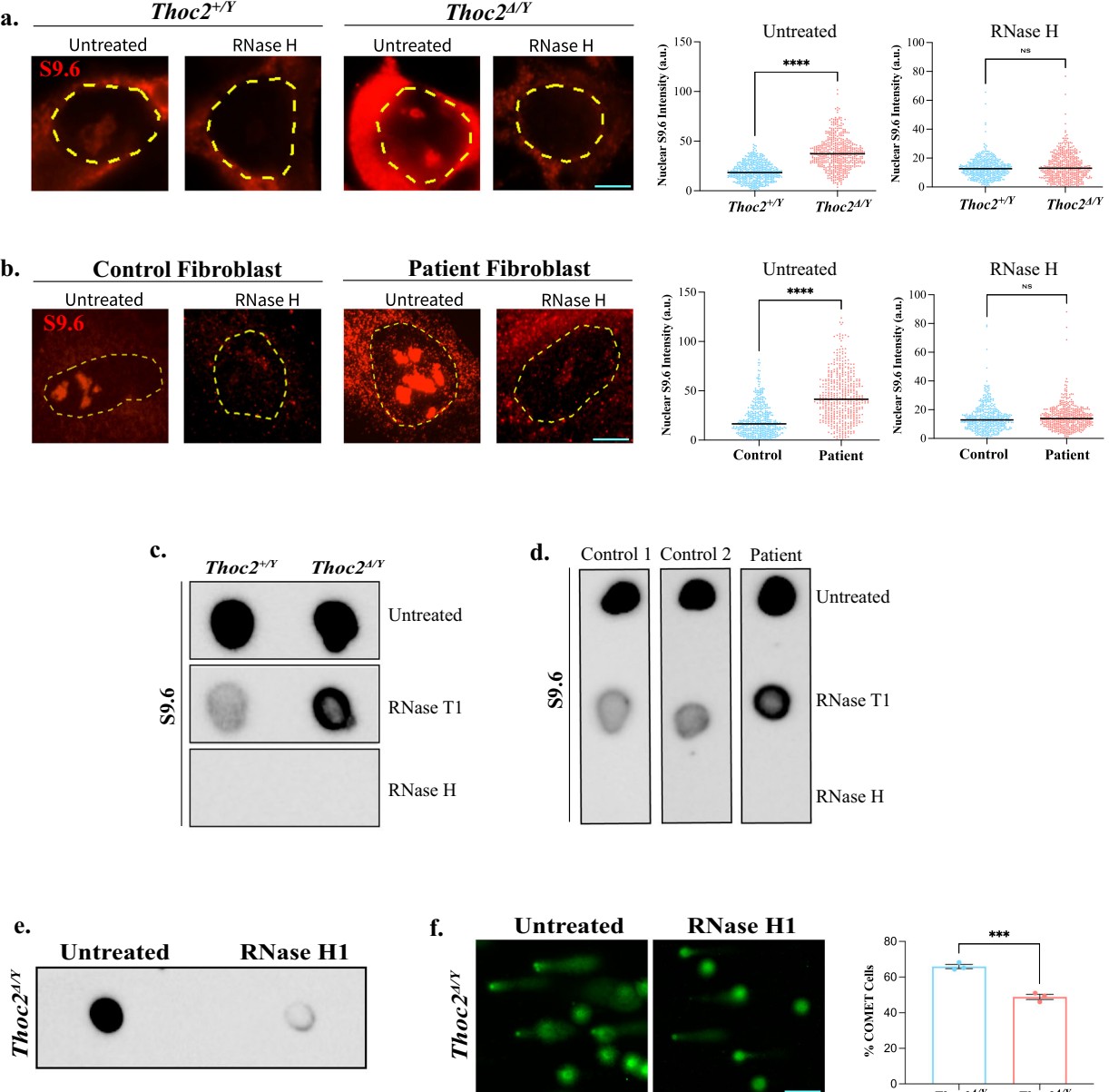

**Fig. 7 | R-loop accumulation and DNA damage in *Thoc2^{Δ/Y}* NSCs and *THOC2* ex37- 38 deletion patient fibroblasts.** Representative immunofluorescent images of untreated or RNase H treated *Thoc2^{+/Y}* and *Thoc2^{Δ/Y}* NSCs (**a**) and control and *THOC2* exon 37-38 deletion patient fibroblasts (**b**) stained with anti-RNA:DNA Hybrid S9.6 antibody (Sigma, MABE1095). Scatter plot showing quantitative measurement of mean nuclear S9.6 fluorescence intensity using Fiji software (at least 300 cells scored per condition over 4 independent experiments); NS non-significant; ****$p < 0.0001$; Mann–Whitney $U$ test with the black line indicating median value. a.u, arbitrary units. Scale Bar 5 μm. **c** Representative images of dot blot assays performed on gDNA from untreated or RNase T1/H treated *Thoc2^{+/Y}* and

*Thoc2^{Δ/Y}* NSCs using anti-RNA:DNA Hybrid S9.6 antibody; $n = 4$ donor embryos per genotype for NSCs. **d** Representative images of dot blot assays performed on gDNA of untreated or RNase T1/H treated control and *THOC2* ex37-38 deletion patient fibroblasts using anti-RNA:DNA Hybrid S9.6 antibody; $n = 3$ independent experiments for control and patient fibroblasts. Representative image of **e**, dot blot assay and **f**, alkaline comet assay, performed on untreated and RNase H1 lentivirus transduced *Thoc2^{Δ/Y}* NSCs (Scale Bar 100 μm). $n = 2$ embryos and 3 independent experiments. Percentage of cells showing comet tail were quantified for **f**; ****$p = 0.0009$; two-tailed unpaired student's $t$ test. Data presented are mean values ± SEM. Source data are provided as a Source Data file.

disordered region[10], we premised that increased DNA damage and transcriptional dysregulation could be due to inefficient resolution of the R-loops in *Thoc2^{Δ/Y}* mice and *THOC2* exon 37-38 deletion patient cells. Indeed, we observed increased accumulation of R-loops in the *Thoc2^{Δ/Y}* NSCs that was significantly reduced on RNase H1 overexpression and was accompanied by a concomitant decline in DNA damage, thus establishing R-loop accumulation as one of the major contributors to the DNA damage detected in *Thoc2^{Δ/Y}* NSCs. We observed similar R-loop accumulation and DNA damage in the *THOC2* exon 37-38 deletion patient-derived fibroblasts, reiterating a

conserved mechanism of perturbed *THOC2* function related cellular anomalies.

Our data clearly showed C-terminally truncated THOC2 protein impacts R-loop homeostasis leading to increased DNA damage and transcriptional dysregulation in *Thoc2^{Δ/Y}* neurons/brain. However, THOC2 transient depletion results in bulk polyA+ mRNA nuclear retention[24]. Still others have shown THOC1, 2, 5 or 6 subunits mediate nuclear export of specific mRNAs, e.g., those involved in pluripotency or haematopoiesis[6,12,21,65,69]. Therefore, we cannot completely rule out an impact on the export of specific mRNAs in the *THOC2* exon 37-38

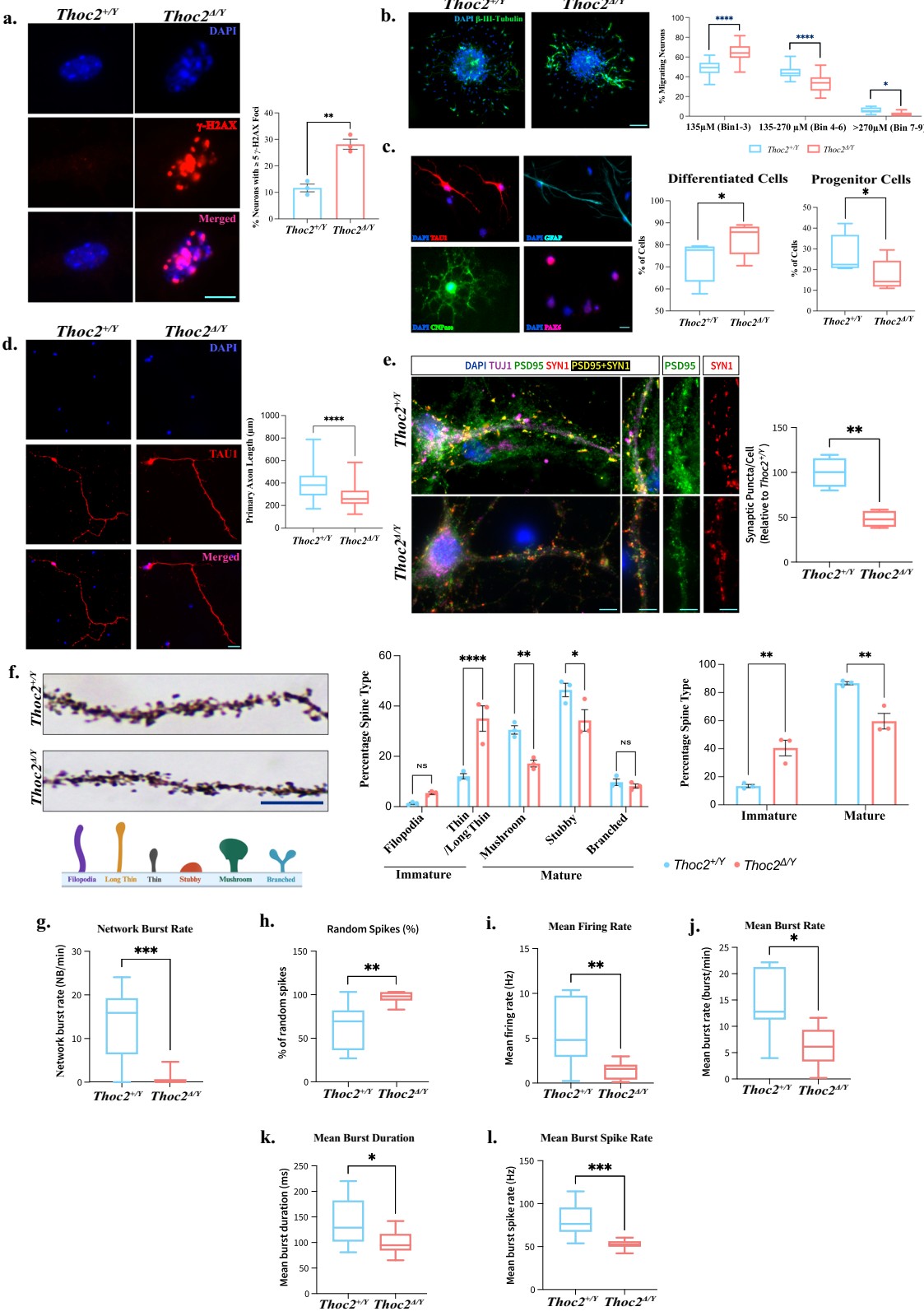

deleted cells even though an obvious nuclear mRNA retention in $Thoc2^{\Delta/Y}$ E18.5 neurons and patient fibroblasts was not observed (see Supplementary Fig. 3a). RNA-seq of neuronal cytoplasmic and nuclear RNA fractions may facilitate identification of a subset of mRNAs whose export from the nucleus may be impacted in $Thoc2^{\Delta/Y}$ neurons and perhaps also in other cell types and stem cell niches. Additionally, THOC2 has been shown as a critical factor for cancer cell survival[59,62,63]

and its mutations are positively correlated with increased survival of cancer patients[67]. Therefore, we believe our $Thoc2$ mouse model can serve as a starting point towards exploration of unprecedented onco-therapeutic strategies.

In summary, our unsuccessful attempts to generate mouse model for p.Leu438Pro or p.Ile800Thr pathogenic patient variants suggested that reduced stability of THOC2 and other THO subunit proteins,

**Fig. 8 | Structural and functional impairments in *Thoc2*^Δ/Y neurons.** Representative immunofluorescent images of **a** nuclei of *Thoc2*^+/Y and *Thoc2*^Δ/Y primary neurons stained with DNA damage marker, γ-H2AX (red) at day in vitro 14 in cultures. Graph showing number of neurons with ≥5 γ-H2AX nuclear foci; *n* = 3 embryos per genotype; **p = 0.0023. **b** Neuronal migration from adhered neurospheres from *Thoc2*^+/Y and *Thoc2*^Δ/Y E18.5 embryonic brains. Neurons were stained using anti-βIII-tubulin antibody (green) and the nuclei with DAPI (blue). Scale Bar: 100 μm; *n* = 4 embryos per genotype; at-least 20 neurospheres scored per replicate; *p = 0.032; ****p < 0.0001. **c** Different cell types (stained with cell specific markers) after 72 h of differentiation in vitro from *Thoc2*^+/Y NSCs are shown. Scale Bar: 50 μm. Percentage of progenitor cells and additive differentiated cells were shown in box plots (*n* = 4 embryos per genotype; at-least 120 cells counted per replicate); *p = 0.04. **d** *Thoc2*^+/Y and *Thoc2*^Δ/Y primary neurons stained with an axonal marker TAU1 (red). Scale Bar: 20 μm. Box plots shows primary axon length; *n* = 4 embryos per genotype; ****p < 0.0001. **e** *Thoc2*^+/Y and *Thoc2*^Δ/Y primary neurons stained with neural βIII-tubulin (purple), postsynaptic PSD95 (green), and presynaptic SYN1(red) markers. PSD95_SYN1 colocalized synaptic puncta are in yellow. Scale Bar: 20 μm. Box plot showing number of excitatory synaptic puncta per neuron; *n* = 4 embryos per genotype; at-least 15 neurons scored per replicate; **p = 0.0033; **a**–**e** two-tailed unpaired student's *t* test, data presented as mean values ± SEM. **f** Left panel: representative images of dendrites from Golgi-Cox stained *Thoc2*^+/Y and *Thoc2*^Δ/Y mice brains (age 30 day) and a schematic diagram showing the different types of spines. Scale Bar: 5 μm. Right panel: graphs showing quantification of different spine types and overall mature and immature spines between *Thoc2*^+/Y and *Thoc2*^Δ/Y mice; *n* = 3 mice per genotype; Total 1852 spines for *Thoc2*^+/Y and 2055 spines for *Thoc2*^Δ/Y mice were analysed: *p = 0.011; **p = 0.0046; ****P = 0.000008; ns: not significant; two-way ANOVA, Bonferroni's multiple comparison test. Quantitative measurement of **g**, network burst rate (***p = 0.0005) and **h**, percentage of random spikes (**p = 0.0023) and **i**–**l**, other neural network parameters; *p = 0.01; **p = 0.002; ***p = 0.0006. **g**–**l**, *n* = 4 embryos per genotype; Kolmogorov–Smirnov test. Source data are provided as a Source Data file. Schematic in **f** was created with BioRender.com.

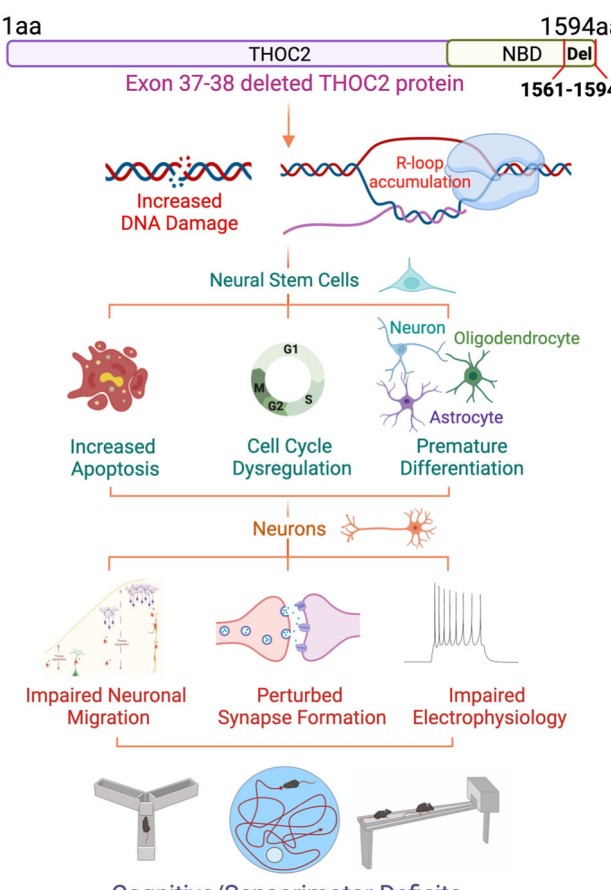

**Fig. 9 | Summary of the key results of studies presented here.** Schematic showing major molecular, cellular, and phenotypic consequences of *Thoc2* exon 37-38 deletion in the *Thoc2*^Δ/Y mice. The figure was created with BioRender.com.

possibly resulting in loss of functional TREX complex, is incompatible for survival in mice. However, the p.Leu438Pro or p.Ile800Thr reduced levels are compatible with life in humans but at the expense of causing NDDs due to altered neuronal development. These observations confirm the importance of *THOC2* in healthy development and differentiation. Following on from this, we successfully generated a viable *Thoc2* mouse model of a unique naturally occurring pathogenic *THOC2* exon 37-38 deletion patient variant. Although THOC2 exon 37-38 truncated protein accumulated and did not impact the levels of other TREX subunits in *Thoc2*^Δ/Y and patient cells, it led to a functionally compromised TREX complex that is sufficient for survival of *Thoc2*^Δ/Y

mice. The *Thoc2*^Δ/Y mice allowed us to investigate biological functions of *THOC2* beyond its essential role in mRNA export. We also found that perturbed R-loop homeostasis and DNA damage contributes to many molecular, cellular, and organism-level anomalies in the *Thoc2*^Δ/Y mice and likely, also in the individuals living with *THOC2* syndrome. Overall, our results establish *Thoc2*^Δ/Y mouse as an invaluable model for deeper understanding of the THOC2/TREX biology in stemness, development and differentiation as well as the clinical neuropathology of *THOC2* syndrome in the future. A schematic diagram showing the molecular, cellular, and phenotypic impact of exon 37-38 deleted truncated *THOC2* protein on *Thoc2*^Δ/Y mice is illustrated in Fig. 9.

## Methods

### Experimental animal model and ethics statement
C57BL/6JArc inbred genetic background mice (www.arc.wa.gov.au) were used for generating the *Thoc2* exon 37-38 deletion mouse model. The mouse colony maintenance and breeding were carried out in Bioresource Facility (BRF), Laboratory Animal Services (LAS), the University of Adelaide. All breeding and experimental procedures were approved by Animal Ethics Committee (AEC) of the University of Adelaide. We abbreviated the *Thoc2* exon 37-38 deletion male mice as *Thoc2*^Δ/Y, wild type (WT) males as *Thoc2*^+/Y, heterozygous females as *Thoc2*^Δ/+, WT females as *Thoc2*^+/+ and homozygous females as *Thoc2*^Δ/Δ. The animals were housed in individually ventilated cages (IVCs) under a 22–24 °C temperature and 40–70% relative humidity environment. The mice were given access to food and water *ad libitum* with a 12 hr light/dark cycle. The *Thoc2*^Δ/Y and *Thoc2*^+/Y mice (at 2 weeks of age and embryonic stages) were genotyped by our optimised protocol that involved rapid genomic DNA extraction (from ear notches of 2-week-old mice or a small amount of embryonic tissue) and multiplex PCR (Supplementary Fig. 1h). We maintained the colony and generated experimental animals by mating *Thoc2*^Δ/+ with *Thoc2*^+/Y mice (Supplementary Fig. 1b).

### CRISPR/Cas9 editing for generating *Thoc2* exon 37-38 deletion mice
Wild type (WT) SpCas9 was complexed with guide RNAs targeting introns 36 and 38 (see Supplementary Data 5) along with PAM killing ssDNA templates (Supplementary Data 5) and then microinjected into the cytoplasm of C57BL/6JArc zygotes. Injected zygotes were then transferred to the oviduct of pseudo-pregnant females for *in-utero* development. The two single-stranded oligos were utilized to insert PAM killing mutations at the intronic cut sites to reduce the proportion of alleles that may be repaired by Non-Homologous End Joining (NHEJ) between the two cut sites which would delete exons 37 and 38 (see Fig. 1a). After successful pregnancy and birth, pups were genotyped using a rapid PCR strategy (Supplementary Fig. 1j) and deletion of Exon 37 and 38 was confirmed by Sanger sequencing. For genotyping the

founder animals, genomic DNA (gDNA) was isolated from a small ear notch (also served as an animal identifier) using High Pure PCR Template Preparation Kit (Roche, 11796828001) as per manufacturer's instructions. PCR was performed with KAPA HiFi HotStart PCR mix (Roche, KK2601) (details in Supplementary Data 5).

## Rapid gDNA extraction and multiplex genotyping PCR

For routine genotyping of the *Thoc2* mice and especially embryos, a rapid gDNA extraction and PCR method was developed. gDNA was isolated from a small ear notch of the *Thoc2* mice (also served as animal identifier) or embryo toe-tip using MyTaq Extract-PCR Kit (Meridian Bioscience, BIO-21127). PCR amplification was performed using MyTaq HS Red mix 2× supplied with the gDNA extraction kit (details in Supplementary Data 5).

## RNA extraction and RT-PCR

RNA was isolated from mouse brain using RNeasy Mini Kit and on-column RNase-free DNase-treatment (Qiagen, 74004). cDNAs were generated from the total RNA (1 μg each) using Superscript IV reverse transcriptase (Invitrogen, 18091050). PCR amplification was performed using Phusion High-Fidelity DNA Polymerase (NEB, M0530S). PCR products were gel purified and subjected to Sanger sequencing.

## In vitro fertilization and blastocyst genotyping

In vitro fertilization was performed to generate large batch of blastocysts using a published protocol[82]. Briefly, the donor females (8–12 weeks old) were first injected with 0.15 ml of pregnant mare's serum gonadotropin (PMSG) (7.5 IU) and 48 h later with 0.15 ml of human chorionic gonadotropin (hCG) (7.5 IU) intraperitoneally. After 15–17 h, the superovulated females were sacrificed by cervical dislocation and oviducts were collected. Similarly, mature donor males (3-4 months old) were sacrificed and cauda epididymides harvested, cut and the clots of spermatozoa were collected in a drop of CARD FERTIUP-PM medium (Cosmo Bio, USA, KYD-002-05-EX) and allowed to capacitate for 60 min by placing the dish in an incubator (37 °C, 5% CO₂). Next, each oviduct ampulla was dissected, and cumulus-oocyte-complexes (COCs) were allowed to release and transferred into drops of CARD Medium (Cosmo Bio, USA, KYD-003-EX). The COCs were then placed in an incubator (37 °C, 5% CO₂) for 30–60 min before starting the in vitro fertilization process. After the incubation period, adequate amount of sperm suspension was added to the drops of COCs. After 3 h of incubation, the oocytes were washed three times using modified human tubal fluid (mHTF)[83] and cultured for further 3 h. After a total of 6 h of incubation with sperm in vitro, presumptive zygotes were assessed for evidence of fertilization (2 pronuclei), divided into groups of 20–30 zygotes per drop of mHTF and cultured in mHTF overnight. Next day, 2-cell embryos were assessed and grouped for further 72-hour culture allowing them to develop to blastocyst stage. The blastocysts were removed from the culture media and transferred into micro-drops of M2 media (Merck, M7167) at room temperature. Individual blastocysts were frozen in PCR tubes in 1 μl of M2 medium plus 9 μl sterile water at -20 °C for subsequent DNA extraction. Blastocyst gDNA was extracted by adding 10 μl of extraction buffer (185.5 mM Tris-HCl pH 8.3, 185.5 mM KCl, 0.008 % gelatin, 0.8 % Tween 20, 1.2 mg/ml baker's yeast tRNA (Sigma, R8759) and 8.5 mg/ml Proteinase K (Thermo Scientific, EO0491) into the individual blastocyst containing PCR tubes followed by incubation at 56 °C for 10 min and then at 95 °C for 10 min. The blastocysts were then genotyped by same the PCR strategy used for genotyping the *Thoc2* embryos and postnatal mice as described above.

## Cell culture

**Fibroblasts.** Dermal fibroblasts from the *THOC2* Ex37-38Del patient and unrelated male controls were maintained in DMEM (Gibco,

10566024) containing 2 mM L-glutamine, 1 mM sodium pyruvate, and 10% foetal bovine serum (FBS) at 37 °C with 5% CO₂.

**Primary Mouse Neurons.** Primary neuronal (cortical and hippocampal) cultures were prepared from E18.5 (18.5 days post coitum) embryonic brains isolated from the time-mated *Thoc2*Δ/+ females as previously described with some modifications[84]. Briefly, the brains were micro-dissected to remove midbrain and the meninges, trypsinised (0.05% w/v) for 20 min at 37 °C with gentle shaking every 5 min. The tissues were then washed with neuron-plating medium (Neurobasal A medium supplemented with 10% FBS, 1 mM sodium pyruvate, 2 mM L-glutamine and 1% penicillin/streptomycin). The cells were triturated using fire-polished glass pipettes and plated in Poly-L-Lysine-coated coverslips (Menzel Gläser, MENZBB018018A2) at a density of $3 \times 10^3$ cells per well of 12-well culture plates for immunofluorescence (IF) studies or $1 \times 10^6$ cells per well of 24-well MEA plate (Multichannel Systems, Germany, 24W300/30G-288) for multielectrode array studies, and maintained in neuron maintenance medium (Neurobasal A supplemented with 2% B-27 supplement, 2 mM Glutamax and 1% penicillin/ streptomycin) and cultured for the required duration with half change of media every 3rd day.

**Primary mouse neural stem cells (NSC).** NSCs were isolated from the dorsal cortex of E18.5 embryonic mouse brains and cultured as non-adherent neurospheres as described[56]. Briefly, dorsal cortex was isolated by micro-dissecting midbrain and the meninges, and then trypsinised (0.05% w/v) for 20 min at 37 °C with gentle shaking every 5 min. The tissues were then washed with NSC medium (DMEM/F12 medium supplemented with 2% B27, 1% penicillin/streptomycin, 20 ng/ml EGF and 20 ng/ml bFGF). The cells were triturated using fire-polished glass pipettes and plated in T25 flasks with NSC medium at a density of $4 \times 10^6$ cells/flask. The neurospheres were supplemented with fresh NSC medium every 3rd Day. For generating the adherent NSCs, passage 2 neurospheres were dissociated into single cell suspension using trypsin (0.05%) in Versene solution (Gibco, 15040066) and plated at a density of $1.8 \times 10^6$ cells/9.5 cm² culture area in NSC medium. The media was half changed every 2nd Day.

## RNase H1 lentivirus generation

Mouse RNase H1 (NM_001286865.1) was PCR amplified from the mouse fibroblast cDNA using attB1-forward and attB2-Reverse primer (sequences in Supplementary Data 5) with Platinum SuperFi PCR Master Mix (Thermo Scientific, 12368010), gel purified and recombined into pDONR221 gateway plasmid. The resulting construct was recombined into pJS65 MND-DEST SV40-Tomato (kindly provided by Prof Richard Iggo, INSERM, Bordeaux, France). The RNase H1 expressing lentivirus was generated by co-transfecting pMD2.G, psPAX2 and pJS65 MND-DEST SV40-Tomato-Myc-mRNase H1 plasmids using published PEI protocol[85]. The lentivirus was tested for RNase H1 expression by transducing the HEK293T cells (ATCC, 293T-CRL-3216) at different MOIs and western blotted with the mouse anti-c-Myc (9E10)-HRP antibody (1:2000) (See Supplementary Data 6) (Supplementary Fig. 8b).

## Western blotting

Mouse tissue samples were lysed in 50 mM Tris−HCl pH 8.0, 150 mM NaCl, 1% IGEPAL CA-630, 0.5% sodium deoxycholate, 0.1% SDS, 1× Protease inhibitor/No EDTA and cultured cells (NSCs and Fibroblasts) in 50 mM Tris−HCl pH 7.5, 250 mM NaCl, 0.1% Triton X-100, 1 mM EDTA, 50 mM NaF, 0.1 mM $Na_3VO_4$ and 1× Protease inhibitor/No EDTA (Roche). The lysates were sonicated using microtip, 15 sec at 30% amplitude (3× times) for tissues and once for 10 sec at 25% amplitude for cells (Sonics Vibra-Cell VCX 130). The tissue/cell debris were removed by centrifugation at 15,000× g for 20 min at 4 °C and proteins assayed using Pierce BCA assay kit (Thermo Scientific, 23227). 8 μg

protein of each sample was resolved on 4-12% Bis-Tris (Bio-Rad, 3450123) or 3–8% Tris-acetate protein gels (Invitrogen, EA03785BOX), transferred to nitrocellulose membranes and western blotted with appropriate primary (1:1000) and HRP conjugated secondary antibodies (1:1000) (see Supplementary Data 6). Chemiluminescence signal was detected by Clarity Western ECL Substrate (Bio-Rad, 1705060) and captured using a Bio-Rad ChemiDoc MP Gel Documentation System. Note: The anti-THOC2-I antibody epitope is located between amino acids 1097-1200 and the anti-THOC2-II antibody between amino acids 1543–1593. The two antibodies were used to distinguish full length and truncated THOC2 proteins from the *Thoc2+/Y* and *Thoc2Δ/Y* mouse brain or patient cell lysates, respectively. The unprocessed western blots are provided in the Source data file.

### H&E staining
E18.5 and P30 mouse brains were fixed in 4% paraformaldehyde (PFA) and embedded in paraffin. 5 μm sections were cut and stained with Mayer's Haematoxylin and Eosin in a Leica Autostainer XL/Leica CV5030 Coverslipper machine following standard protocols. Images were acquired using 3D Histech Pannoramic Scan II machine equipped with Carl Zeiss Plan-Apochromat ×20 objective with Point Grey Grasshopper 3 CCD monochrome camera and LED-based RGB illumination unit. The image measurements and annotations were performed in the 3D Histech SlideViewer (V2.6) software.

### Golgi-Cox staining
Golgi-cox staining on P30 mouse brain was performed using a previously reported protocol with slight modification[86]. Briefly, mouse brains were perfused using 4% PFA, removed from the skull and immersed in 4% PFA overnight at 4°C. The brains were then incubated for 72 h in Golgi-Cox stain (5% potassium dichromate solution, 5% mercuric chloride and 5% potassium chromate) at 37 °C. The tissue blocks were then removed and placed into 30% sucrose solution for 3 days at room temperature. Sections (200 μm thick) were cut on a vibratome, and images were taken using an Olympus IX83 microscope (Olympus, Japan). Subsequently, analysis of the dendritic spines was carried out using Fiji software (NIH). All sample analysis was carried out blind to mouse genotype.

### Immunofluorescence (IF)
**Cultured cells.** NSCs, primary mouse neurons and fibroblasts were cultured on Poly-L-Lysine (Sigma, P2636) coated glass coverslips and immunostained using a modified protocol reported earlier[3]. Briefly, cells were fixed with 4% PFA for 10 min, washed three times with 1× PBS, permeabilized with 0.2% Triton X-100 in 1× PBS for 10 min, washed three times with 1× PBS, and blocked in 10% Horse Serum in 1× PBS for 1 h at room temperature. Cells were then incubated with appropriate primary antibodies (1:1000) (see Supplementary Data 6) diluted in 10% Horse Serum in 1× PBS overnight at 4 °C. The following day, cells were washed three times with 1× PBS, incubated with appropriate Alexa Fluor secondary antibodies (1:1500) (see Supplementary Data 6) for 2 h at room temperature, washed three times with 1× PBS and then mounted in ProLong Gold Antifade Mounting agent with DAPI (Invitrogen, P36935) with overnight curing.

**Brain sections.** The IF protocol for tissue sections was used as described earlier[57] with some modifications. Briefly, E18.5 embryonic mouse brains were fixed by immersion in 4% PFA for 24 h, cryoprotected in 30% sucrose and embedded in Optimal cutting temperature compound (OCT, Tissue-Tek, IA018). A Leica CM1900 cryostat was used to prepare 16 μm sections that were permeabilized using 0.5% Triton X-100 in 1× PBS for 5 min at room temperature. The sections were washed three times with 1× PBS, blocked with 10% Horse Serum, 1% BSA and 0.1% Triton X-100 in 1× PBS for 1 hr at room temperature and incubated with appropriate primary antibodies (1:500 dilution)

(see Supplementary Data 6) diluted in 10% Horse Serum, 1% BSA and 0.1% Triton X-100 in 1× PBS overnight at 4 °C. Following day, sections were washed three times with 0.05% Triton X-100 in 1× PBS, incubated with appropriate Alexa Fluor secondary antibodies (1:1000 dilution) (see Supplementary Data 6) for 3-4hrs at room temperature, washed three times with 1× PBS and mounted in ProLong Gold Antifade Mounting agent with DAPI with overnight curing.

### RNA-DNA hybrid immunofluorescence using S9.6 antibody.
S9.6 IF was performed as per published protocols[87,88]. Briefly, mouse NSCs and patient fibroblasts were fixed in ice-cold 100% methanol for 10 min at −20 °C and blocked using 1% BSA in 1× PBS for 1 h at room temperature. Cells were treated with RNase T1 (Thermo Scientific, EN0541) or RNase H (NEB, M0297S) in the presence of appropriate buffers supplied with the enzymes and then incubated with mouse anti-S9.6 antibody (Sigma, MABE1095 or Kerafast, AB01137-23.0) at 1:1000 in 1% BSA in 1× PBS overnight at 4 °C. Next day, cells were washed three times with 1× PBS, incubated with secondary anti-rabbit Alexa Fluor 555 conjugate (see Supplementary Data 6) diluted in 1× PBS with 1% BSA (1:2000) for 2 h at room temperature and finally, mounted in ProLong Gold Antifade Mounting agent with DAPI. Images were captured using a Zeiss Axioplan2 microscope (Carl Zeiss, Jena, Germany) equipped with an HBO 100 lamp (Carl Zeiss) using Axiocam Mrm camera and V.4.9.1.0 software (Axiovision, Carl Zeiss) at 63×63 magnification. For NSCs, image acquisition settings were calibrated using the *Thoc2+/Y* NSCs, and for patient fibroblasts, image acquisition settings were calibrated using unrelated control fibroblasts. The same settings were then applied to assess the experimental samples i.e., *Thoc2Δ/Y* NSCs and *THOC2* exon 37-38 deletion patient fibroblasts. Fiji (ImageJ) software was used for measuring mean nuclear grey value for S9.6 fluorescence[89].

### Fluorescence in situ hybridization (FISH)
FISH was used to detect polyA+ RNA in human fibroblasts and mouse neurons as described previously[90]. Briefly, cells were washed once with 1× PBS, fixed with 4% PFA for 10 min, washed three times with 1× PBS, and then permeabilized with 0.5% Triton X-100 in 1× PBS for 5 min, washed two times with 1× PBS and then once with 2× SSC for 10 mins at room temperature. An oligo dT (70) probe (HPLC purified) labelled with Alexa Fluor 546 at the 5' end, was then added at a concentration of 1 ng/μl and incubation was performed overnight at 42 °C. The cells were washed two times with 2× SSC, for 15 min each, then once with 0.5× SSC for 5 min, and lastly, once with 1× PBS for 5 min at room temperature. Images were captured using a Zeiss Axioplan2 microscope (Carl Zeiss, Jena, Germany) equipped with an HBO 100 lamp (Carl Zeiss) using Axiocam Mrm camera and V.4.9.1.0 software (Axiovision, Carl Zeiss).

### Flow cytometry
**Apoptosis assay.** Apoptotic NSCs were detected by annexin V staining using a published protocol[91] with some modifications. Briefly, NSCs were harvested, washed in 1× PBS, and resuspended in 20 μl annexin V binding buffer (Hank's balanced salt solution with 1% HEPES and 5 mmol/L CaCl$_2$) with 0.075 μg/ml annexin V-PE (Stem Cell Technologies, 100-0331) at 1 × 10$^5$ cells/ml. The cells were stained for 20 min at 4 °C, diluted with 200 μl of ice-cold annexin V binding buffer, and analysed immediately on a LSRFortessa X20 Flow cytometer (BD Biosciences).

**Cell cycle assay.** For cell cycle assay, NSCs were resuspended in pre-warmed media at 1 × 10$^6$ cells/ml. Hoechst 33342 (Sigma, B2261) added at a final concentration of 10 μg/ml, mixed thoroughly, and incubated at 37 °C for 45 min in dark. The cells were then pelleted and resuspended in 500 μL of cold PFE solution (1× PBS, 2% FBS, 2 mM EDTA) and analysed on a LSRFortessa X20 Flow cytometer (BD Biosciences).

All Flow Cytometry data were analysed using FlowJo V. 10.0.8 software (FlowJo, LLC). The gating strategy has been exemplified in Supplementary Fig. 9 provided in the Supplementary information file.

## TUNEL assay

TUNEL assay was performed on NSCs cultured on Poly-L-Lysine treated coverslips using Click-iT Plus TUNEL Assay kit (Invitrogen, C10617). The coverslips were then mounted using ProLong Gold Antifade Mounting agent with DAPI and imaged on a Zeiss Axioplan2 microscope using Axiocam Mrm camera and V.4.9.1.0 software at ×20 magnification. Percentage of apoptotic NSCs was determined by quantifying TUNEL-positive cells in five random microscopic fields capturing ≥100 cells per technical replicate.

## Single cell electrophoresis assay

DNA damage in E18.5 mouse NSCs and patient fibroblasts was assessed using alkaline single cell electrophoresis or comet assay[47] with minor modifications. Briefly, NSCs were harvested and resuspended in 1% low-melting point agarose prepared in distilled water. The cell suspension was dropped on glass slides precoated with a thin layer of 1% agarose in distilled water and spread with a coverslip (22 mm × 22 mm) to make the electrophoresis mini gel. The cells were lysed using comet lysis buffer (2.5 M NaCl, 100 mM disodium EDTA, 10 mM Tris-HCl pH 10, 200 mM NaOH, 1% Triton X-100 and 1% sodium lauryl sarcosinate) for 1 hr at 4 °C. Electrophoresis was performed using an alkaline electrophoresis solution (200 mM NaOH, 1 mM disodium EDTA, pH >13). The slides were stained with SYBRGold Nucleic Acid Gel Stain (Invitrogen, S11494) and imaged using a Zeiss Axioplan2 microscope. The tail moments were quantified using the CaspLab software[92].

## Dot blot assay

Dot blot assay was performed using a published protocol[64] with some modifications. Briefly, gDNA was isolated from mouse NSCs or patient fibroblasts using DNeasy Blood and Tissue Kit (Qiagen, 69504) without the RNase treatment. Equal amount of gDNA of each sample (2 μg) was treated with RNase T1 (Thermo Scientific, EN0541), RNase H (NEB, M0297S) or buffer at 37 °C for 1 hr. The samples were spotted on a nitrocellulose membrane, cross-linked using a UV cross linker (GS Gene Linker, Bio-Rad) (120 mJ) and blocked in 5% milk in PBST (1× PBS, 0.1% Tween 20) overnight. The membrane was incubated with rabbit S9.6 antibody (1:1000) diluted in 3% milk, 1% BSA in PBST overnight at 4 °C. Next day, the membrane was washed three times with PBST, incubated with goat anti-rabbit-HRP antibody (Agilent, P0448) diluted in 3% milk, 1% BSA in PBST (1:1000) for 2 h at room temperature, washed three times with PBST and visualized using Clarity Western ECL Substrate on a Bio-Rad ChemiDoc MP Gel Documentation System.

## Neural migration assay

In vitro neuronal migration assay was performed as reported[54]. Neurospheres at Passage 3 were cultured for 5 days and then seeded on Poly-L-Lysine coated coverslips at 100 neurospheres/9.5 cm² in NSC medium without the growth factors. Neurons were allowed to migrate away from the seeded neurosphere for 48 h and then fixed with 4% PFA. Immunofluorescence was performed using anti-ß-Tubulin III (Tuj1) antibody (1:1000 dilution). A neuronal migration scoring system we published was used[52]. Neurospheres and the migrating neurons were imaged using a Zeiss Axioplan2 microscope (Carl Zeiss). The edge of contiguous nuclei was used as neurosphere periphery and its outline was used to generate concentric circles (bins) using Fiji software, outside of the neurosphere (which are the migration zones) with increasing size (each bin increases by 45 μm). Neural cell density was then calculated in each bin, which results in migration distance measurement independent of initial sphere size, and bin volumes.

## Neural differentiation assay

Passage 3 neurospheres were cultured for 5 days, dissociated into single cell suspension, and plated on Poly-L-Lysine coated coverslips at 2.1 × 10⁴ cells/9.5 cm² culture area in NSC medium without the growth factors. The cells were allowed to differentiate for 72 h and then fixed with 4% PFA. Immunofluorescence was performed as above. Cell type specific primary antibodies used were: GFAP (for astrocytes), CNPASE (for oligodendrocytes), ß-III TUBULIN (for neurons), and PAX6 (for progenitors). The details of the primary (all diluted 1:1000) and secondary antibodies (all diluted 1:2000) are provided in the Supplementary Data 6.

## Multielectrode array analysis

Spontaneous neural network activity was recorded using 24-well multielectrode arrays (MEAs) (Multichannel systems, MCS GmbH, Reutlingen, Germany) embedded with 12 gold electrodes with a diameter of 30 μm, spaced 300 μm apart as described[57] with some modifications. The wells were coated with Poly-L-Lysine and neurons were plated at 1 × 10⁶ cells/well. Before recording neuronal activity, the plates were equilibrated in the MEA recording chamber for 30 min at 37 °C with 5% CO₂. Neuronal activity was then recorded for 15 min. Recordings were sampled at 20,000 Hz and filtered using a high-pass 2nd order Butterworth filter with a 100 Hz cut-off, and a low-pass 4th order Butterworth filter with a 3500 Hz cut-off. The individual spike detection threshold was set at ±6.0 standard deviations. MEA data was analyzed using Multiwell analyzer software (Multichannel systems) and a MATLAB (The Mathworks, Natick, MA, USA) code to extract parameters describing the spontaneous network activity[93]. The mean firing rate (MFR) (Hz) was calculated for each well individually by averaging the firing rate of each separate electrode by the total amount of active electrodes of the well. Bursts were detected by the Multiwell analyzer built-in burst detection algorithm. Bursts were defined with a maximum of 50 ms inter spike interval (ISI) to start a burst, a maximum of 50 ms ISI to end a burst, with a minimum of 4 spikes per bursts, and a minimum of 100 ms inter burst interval (IBI). Synchronous network bursts were detected when a burst was present in at least 50% of all electrodes. The percentage of random spikes (PRS) was calculated using the percentage of isolated spikes that did not belong to a burst. The IBI, and network burst IBI (NIBI) were calculated by subtraction of the time stamp of the beginning from the time stamp of the ending of each burst or network burst.

## Synaptic puncta analysis

Quantification of excitatory synapses was performed as reported[57]. Neurons were stained with dendritic MAP2 (purple) (1:1000), presynaptic SYN1 (red) (1:500) and postsynaptic PSD95 (green) (1:500) markers and imaged using Zeiss Axioplan2 microscope (Carl Zeiss) at ×63 magnification. Fiji software (NIH) was used for quantitative analysis of SYN1/PSD95 co-localisation puncta (yellow) by manual labelling using the multi-point tool. Antibody details are provided in the Supplementary Data 6.

## Whole genome sequencing and analysis

gDNA was isolated using High Pure PCR Template Preparation Kit (Roche, 11796828001) as per manufacturer's instructions. gDNA of each sample was initially quantified by NanoDrop (Thermo Fisher Scientific Inc.) and quality checked for purity and integrity by agarose gel electrophoresis, followed by final quantification using Qubit 2.0 fluorometer. Next generation sequencing libraries were prepared using NEBNext Ultra II DNA Library Prep Kit for Illumina with NEBNext Multiplex Oligos for Illumina (NEB E7645). All library clean-up steps were performed using Agencourt AMPure XP (Beckman Coulter). Paired-End 150 bp sequencing was performed using an Illumina NovaSeq 6000 system according to manufacturer's instructions. Sequence reads were mapped to GRCm38 with BWA-MEM (0.7.15) and

variants were called using GATKv4.1.6. All variants were annotated for allele frequency, locus identity and likely pathogenicity using SnpEff[94] and filtered for significance using SnpSift[95]. The copy number variations were detected and characterized using CNVnator[96]. The alignments for the sequence reads were viewed with the Integrative Genomics Viewer (IGV) v.2.14.0.

### RNA sequencing and analysis

Total RNAs were extracted from the E14.5, E18.5 and P10 mouse brain dorsal cortex (we refer to it as 'brain' in the text for simplicity) by in-house protocol that uses combination of TRIzol Reagent (Invitrogen) and RNeasy Mini Kit (Qiagen) and quantified and qualified by Nano-Drop (Thermo Fisher Scientific Inc.), Agilent 2100 Bioanalyzer (Agilent Technologies, USA), and 1% agarose gel. 1 µg total RNA from each sample with RNA integrity number (RIN) value ≥ 7 was used for library preparation. Next generation sequencing libraries were prepared using VAHTS Universal V8 RNA-seq Library Prep Kit for Illumina as per manufacturer's protocol (Vazyme Biotech). Briefly, the poly(A) mRNAs were isolated using NEBNext Poly(A) mRNA Magnetic Isolation Module (NEB). The mRNA fragmentation and priming were performed using NEBNext First Strand Synthesis Reaction Buffer and Random Primer mix. The first strand cDNA was synthesized using NEB ProtoScript II Reverse Transcriptase and the second-strand cDNA using Second Strand Synthesis Enzyme Mix. The double-stranded cDNAs were purified using AxyPrep Mag PCR Clean-up (Axygen) and treated with NEBNext Ultra End Repair/dA-Tailing mix to convert fragmented DNA to repaired DNA and add dA-tails to both ends, followed by T-A ligation using NEBNxt Ultra II Ligation Module to add adaptors to both the ends. The adaptor-ligated DNAs were size selected for fragments of ~360 bp (with ~300 bp inserts) using AxyPrep Mag PCR Clean-Up kit (Axygen). Each sample was amplified by PCR for 11 cycles using P5 and P7 primers; both primers carried sequences that can anneal with flow cell to perform bridge PCR and P7 primer with a six-base index allowing for multiplexing. The PCR products were cleaned up using AxyPrep Mag PCR Clean-Up Kit (Axygen), validated using an Agilent 2100 Bioanalyzer (Agilent Technologies, USA), and quantified by Qubit 2.0 Fluorometer (Invitrogen, USA). The libraries with different indices were multiplexed and loaded on an Illumina NovaSeq 6000 (Illumina, USA). Strand-specific sequencing at 80 million 2 × 150 bp paired-end reads per sample was carried out. Sequence read pairs from each sample were quantified against the mouse transcripts using Salmon v1.6.0[97] with prebuilt indexes for the mouse mm10 genome build available from RefGenie (http://refgenomes.databio.org/v3/assets/splash/0f10d83b1050c08dd53189986f60970b92a315aa7a16a6f1/salmon_sa_index?tag=default)[98]. Differential gene and transcript expression between *Thoc2*$^{+/Y}$ and *Thoc2*$^{Δ/Y}$ genotypes were assessed at each developmental time point separately (E14.5, E18.5 and P10) using the default options for the *glmQLFit* function on transcripts that were included after using the *filterByExpr* function from the *edgeR* v 3.40.2 package in the R v 4.2.2 environment[99]. Lists of expressed genes or transcripts not excluded by the *filterByExpr* function at each time point were used as the background in subsequent gene-ontology analyses. Gene ontology term enrichments were analysed using the g:GOSt functional profiling tool available from the g:Profiler web service (https://biit.cs.ut.ee/gprofiler/gost)[39]. ENSEMBL mouse gene ID were submitted as an ordered query ranked by ascending p-values to a cut off value < 0.01 for each time point. Significantly enriched terms (FDR < 0.05) identified from g:GOSt analysis were then clustered using the REVIGO web service (http://revigo.irb.hr/)[100]. The percent GC content of the differentially expressed genes were compared to the mouse genome using ShinyGO v0.76[101].

### Total brain proteome analysis using LC-MS

Embryonic day 18.5 and adult Day 10 mouse brains were lysed in 500 µl 0.1 M triethylammonium bicarbonate (TEAB) using the Precellys tube homogeniser. The cell lysates were heat-treated for 5 min at 95 °C and further homogenized using a sonicator probe at 4 °C (80% output). Samples were centrifuged at 13,000 × g for 10 min at 4 °C to remove cell debris. Protein concentration in the supernatants was estimated using the Nanodrop (Thermo Fisher Scientific). A total of 100 µg protein from each sample was reduced using 10 mM tris(2-carboxyethyl) phosphine (TCEP) with 0.05% SDS, alkylated with 200 mM methyl methanethiosulfonate (Sigma Aldrich, Australia) and digested overnight with 50:1 trypsin (Pierce Thermo Fischer Scientific, Australia) at 37 °C. Trypsin digestion was stopped by incubation with 10% tri-fluoroacetic acid (TFA) at 37 °C for 15 min followed by centrifugation at 13,000 × g for 10 min at 4 °C The supernatants containing digested peptides were dried down and dissolved in 50 µl 0.1% trifluoroacetic acid and desalted using home-made POROS R3 (Life Technologies, Mulgrave, Victoria, Australia) micro-column. The clean peptides were then used for LC-MS/MS.

The peptide samples (200 ng per sample) were separated by nano-liquid chromatography (nLC) using the Dionex RSLC 3500 system (Thermo Fischer, Australia) coupled online to the trapped-ion mobility spectrometry Time-of-Flight (timsTOF) *Pro* mass spectrometer (Bruker Daltonics, Germany) using the Data Independent Acquisition-Parallel Accumulation Serial Fragmentation (DIA-PASEF) mode. Reversed-phase chromatography was done using a 25 cm, 75 µm ID Aurora C18 nano column (IonOpticks, Australia) with an integrated captive spray emitter. The peptides were eluted using a 125-minute gradient from 0% to 35% buffer B (acetonitrile in 0.1% formic acid) at a rate of 400 nL/minute. Buffer A consisted of 0.1% (v/v) formic acid.

The raw data files from each sample generated on the timsTOF Pro mass spectrometer were processed using the software package Spectronaut™ v16.3 (Biognosys)[102]. directDIA™ analysis was performed and the data were searched against the Uniprot *Mus musculus* FASTA database (17,056 entries; Year 2021). The following additional parameters were applied: variable modifications–deamidation (N/Q), oxidation (M); fixed modification–methylthio (C); enzyme–Trypsin/P; missed cleavages–2. All other parameters were set to default values. Proteins with a false discovery rate (FDR) of ≤1% were reported. Gene Ontology (GO) enrichment analysis was performed using Enrichr web-platform[103,104]. For the Enricher plots, p value values for function clustering and multiple detection corrections were calculated, and the plots were sorted based on the Benjamini-Hochberg corrected p value represented as the length of the GO term bars. Volcano plots were generated using R-studio[105]. Venn diagrams were generated using jvenn online platform[106]. The percent GC content of the genes corresponding to dysregulated proteins were compared to the mouse genome using ShinyGO v0.76[101].

### Behavioural testing

**Morris water maze.** This test was used for assessing spatial and related forms of learning and memory. The 1.2 m diameter water maze pool was filled to a depth of 40 cm with 22 °C water, leaving 15 cm diameter submerged platform 1 cm below the water level. Approximately 500 ml of non-toxic white paint was added to the water to make it opaque and hide the platform from the animal. It also served as a contrast agent for tracking the dark-coloured mice. The mice were placed in the water (one at a time) at one of the cardinal points: North (N), South (S), East (E) and West (W) that created imaginary four equal quadrants and allowed 2 min to find the platform. If the mouse found the platform within this time it was allowed 10 sec on the platform before being removed, gently towelled down, and placed under a warming lamp. If the mouse failed to find the platform within the 2 min time window, it was guided to the platform and left for 10 sec on the platform before being removed, towelled down, and placed under a warming lamp. The routine was repeated four times per day, with an interval time of 20–30 min between swims, for 5 days. A probe trial was run at the end of reference memory trials to assess working memory.

Video capture and quantification was performed using the CleverSys TopScan tracking software.

**Y-maze.** The Y-Maze (SD instruments), used to assess willingness to explore new environment, had three identical arms of 30 cm in length, which were symmetrical to each other. The walls of the apparatus were 14 cm high. For the first trial, the mouse was placed at the end of the home arm, facing away from the centre, and allowed to explore 2 of the 3 Y-Maze arms for a 10-minute period. A partition blocking off the novel arm of the maze was in place during this initial trial (Trial 1). Each of the three arms were marked by a unique cue attached to the end to differentiate it from the others. The cues used in the present study were 2-dimensional pictures of black and white symbols including a circle, stripes, and triangles. Two hours later, the test was repeated with the partition removed so that all arms, including the novel arm, were available to explore (Trial 2). The time spent in each of the three accessible arms was video recorded during 5-minute trial, and the data, including arm entries and time spent in each arm, were electronically recorded. Analysis was performed using TopScan rodent tracking software. Mice typically prefer to investigate the novel arm in the maze rather than returning to one that has been previously visited. Therefore, mice with intact working memory should remember not having visited the novel arm and should spend a greater proportion of time exploring that arm relative to the other arms, a parameter termed as spontaneous alteration and utilized for assessing the working memory of mice.

**Barnes maze.** The Barnes maze test, used for testing spatial learning, spatial reference memory and executive function, was performed using a modified method[107]. The maze is an elevated, open circular platform with holes evenly distributed around the periphery. One of the holes is pre-determined as the escape hole, with a black escape box placed below the hole. The test was performed over the course of five days: three days of acquisition trials, a rest day, and a probe day. During the trials, animals were subjected to two trials spaced 15–20 min apart. Mice were placed in the centre of the Barnes maze in a brightly lit room and the time taken by the animals to find and enter the escape box recorded. On day 5, two trials were conducted 30 min apart. In both trials, the time taken to locate and enter both the location of the old escape box, as well as a novel escape box location, were recorded. Data was recorded using the ANY-maze Video Tracking System version 4.99 m (Stoelting Co.).

**Rotarod test.** This test was used for testing balance, grip-strength, and motor coordination. Mice were pre-trained on the rotarod (Ugo Basile, Italy) for three 5-minute trials. The first two trials were run at a constant speed of 4 rpm, and the 3rd one at an accelerating speed ranging from 4–40 rpm over the 5-minute period. The testing phase was carried out on the next day and involved three trials. Each trial involved the accelerating setting (4–40 rpm) across the 5-minute trial. The time a mouse stayed on the rotating rod before falling was taken as a measure of motor co-ordination. The interval between trials was 25–30 min.

**DigiGait test.** This test was used for in-depth assessment of gait. The mouse was placed inside the testing chamber positioned above a clear Perspex treadmill belt. The treadmill was started at a low speed (usually 10 cm/sec) until the animal was comfortable with walking on the treadmill. The speed was then slowly increased up to 25 cm/sec (depending on the motor functionality of the mice) and recordings of ventral view of mice were made using a high-speed camera positioned below. Approximately 15 steps were captured. The mouse was then returned to its home cage, with the instrument automatically computing the parameters of interest, such as Swing duration of the paws, Braking time, Propulsion time, and Stride length.

**Open field test.** The open field test was used to assess locomotor activity and anxiety, based on the innate but conflicting tendencies of mice to avoid bright light, and open spaces and engage in the exploration of a novel space. Mice were placed into a brightly lit open field (1 m × 1 m, grey arena) for 10 min. Open field activity was recorded and analysed using CleverSys tracking software. Anxiety-like behaviour was assessed by examining the time that mice spent in the perimeter relative to the centre of the maze.

**Beam walking.** This test was used for assessing fine motor coordination and balance in mice. A black acrylic beam 1 m in length and tapering in width from 3.5 cm to 0.5 cm was elevated ~50 cm above a padded cushion. Animals were placed on the wider end of the beam and required to walk up the incline to the narrow end, where an enclosed box was positioned. Animals underwent a training comprised of 3 trials at least 48 hr before testing. Mice were subjected to three consecutive test trials with a 30-minute interval between trials. The time to traverse the beam was recorded and test trials were filmed on both sides from the escape box end for retrospective analysis of foot slips. Recordings were analysed by a single observer blinded to the treatment condition. The results were averaged between the three test trials.

**Pasta manipulation (Capellini Handling Test).** The test quantitatively measures forepaw dexterity and was performed as reported[37]. In brief, capellini pasta pieces (1.0 mm diameter) were cut to 2.6 cm lengths and marked at 0.86 cm increments to allow for visualisation of pasta strand movement during testing. Animals were given capellini pasta in their home cage to eat for 7 days prior to the test and were simultaneously habituated in the plexiglass testing chamber for up 10 min each day. On the day prior to test day, food was removed from the home cage overnight (~10–12 hrs). On the test day, mice were placed in the Plexiglass cage, on top of and surrounded by mirrors, for 5 min. Video cameras, with a high frame rate and ×16 zoom lens, were set up on both sides to capture the reflection of the paws of mice in the mirror while they eat/manipulate the pasta. After 5 min of habituation, a piece of pasta was dropped into the cage and recording continued until a maximum of 3 pasta were consumed. The trial length was 10 min. Parameters such as average eating time per pasta, no of times pasta dropped, paw adjustments while eating and atypical behaviors were analysed from the captured video.

**Adhesive removal test.** This test was performed as described earlier[38] to assess sensorimotor function in the mice. During the experiments, a piece of adhesive tape (3 mm × 3 mm) was gently placed on the forepaw by using blunt-end forceps. Sensorimotor performance was investigated by measuring the latency to contact the adhesive (time-to-contact) and the latency to remove the adhesive from each paw (time-to-remove). The maximum trial length was 120 sec. Trials were conducted one paw per trial, two trials for each paw, for a total of four trials, and latency was recorded and averaged across the trials.

**Elevated plus maze.** The elevated plus maze test was used to measure anxiety. The maze was comprised of two open arms and two closed arms that extended from a common central platform elevated to a height of 40 cm above the floor. The closed arms provided protection via a 15 cm high wall with a passageway 4.5 cm wide. The open arms had no walls but were of the same width as the closed arms. Mice were placed on the centre square, facing an open arm, and allowed to freely explore the apparatus for a 10-minute period. Video capture and analysis for parameters including number of arm entries and time spent in the closed versus open arms was performed using the CleverSys TopScan tracking software.

**Light–dark test**. The light-dark test measures anxiety-like behaviour and was conducted in the TruScan locomotor system (Coulbourn Instruments, USA). The locomotor cells were 40.6 cm wide 40.6 cm deep × 40.6 cm high, encased by twin photo-optic arrays (sensor rings), set up to quantify and qualify movements in the horizontal plane. A black Perspex insert was used to create an area in which half the cell is dark, and half is light (750 lux). The mice were placed in the dark side and left to wander freely for 10 min. The photocells in the sensor rings measure movement from the dark side to the light side and back, and the amount of time spent in each zone was calculated by the TruScan software. The light/dark exploration test measures the conflict between the natural tendencies of a mouse to explore a novel environment versus their tendency to avoid a brightly lit open field. More anxious animals spend less time in the lit area.

**Grip strength test**. A grip strength meter (Ugo Basile, Italy) was used to test the forelimb grip strength of the mice. The technique relies on the inherent resistance of mice to a backward movement due to which they tend to grasp/grip any available object to counter against the pull-back applied by the researcher. The mouse (one at a time) was moved close to the T-shaped bar with its torso horizontal and only forepaws touching the bar. The mouse was then gently pulled backwards by its tail making sure that it gripped the frontal portion of the T-bar. The digital meter recorded the peak force applied by the mouse while grasping the T-bar. Three trials were performed per animal to generate the average grip strength measurement data.

### Statistics and reproducibility
All results are presented as the mean values ± SEM with details of experimental and biological replicates provided in the figure legends. Comparisons between two groups were analysed by Student's $t$ test and multiple-groups by one/two-way ANOVA using GraphPad Prism 9. In all box plots, the boundary of the box closest to zero indicates the 25th percentile and the boundary farthest from zero indicates the 75th percentile. A line within the box shows the median, and the whiskers above and below the box indicate the maximum and minimum points of the dataset, respectively.

### Reporting summary
Further information on research design is available in the Nature Portfolio Reporting Summary linked to this article.

## Data availability
The whole genome sequencing and RNA-Seq data generated in this study have been deposited in NCBI's Gene Expression Omnibus and are accessible through GEO Series accession number GSE245539. The proteomics data generated in this study have been deposited to the ProteomeXchange Consortium (http://proteomecentral.proteomexchange.org) via the PRIDE partner repository with the dataset identifier PXD040358. All remaining data supporting the findings in this study are provided in the manuscript files and/or the Supplementary Information files. Source data are provided with this paper.

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

## Acknowledgements

We would like thank Lottie Servin, Rosanna Duncan, and team from Laboratory Animal Services, The University of Adelaide for helping with mouse colony maintenance, breeding, and health monitoring. Prof. Richard Iggo, INSERM, Bordeaux, France, for his kind gift of pJS65 MND-DEST SV40-Tomato plasmid. Agatha Labrinidis and Jane Sibbons, Adelaide Microscopy, The University of Adelaide, for assisting with microscopy services. Giulia Parodi, Sergio Martinoia's Bioengineering lab, University of Genova, for generating the raster plots from MEA results. Shreya Agarwala, The University of Adelaide, for generating illustrations of behavioural testing. Tina Cardamone and Aira Nuguid, Phenomics Australia, University of Melbourne, for providing Histopathology and Slide Scanning Service. A/Prof. John Finnie from discipline of Anatomy and Pathology, School of Medicine, University of Adelaide, for helping

with histopathology reports. The authors acknowledge the facilities and the scientific and technical assistance of the South Australian Genome Editing (SAGE) Facility, the University of Adelaide, and the South Australian Health and Medical Research Institute. SAGE is supported by Phenomics Australia. Phenomics Australia is supported by the Australian Government through the National Collaborative Research Infrastructure Strategy (NCRIS) program. G.B. is supported by the Angela Wright Bennett Foundation, The McCusker Charitable Foundation via Channel 7 Telethon Trust, The Stan Perron Charitable Foundation and Mineral Resources Limited. J.E.N. was supported by the Veronika Sacco Clinical Cancer Research Fellowship from the Florey Medical Research Foundation. This study was financially supported by the National Health and Medical Research Council of Australia (NHMRC) Fellowship APP1155224 to J.G. and NHMRC project grant APP1163240 to J.G., R.K., and P.Q.T.

## Author contributions

R.B., R.K., and J.G. designed the study and experiments. R.B. performed the majority of the experiments and analyses with substantial input from R.K. L.A.J. advised on the design and execution of primary neural stem cell and differentiated neuron studies. M.A.C. analysed the RNA-Seq data and helped with its presentation. R.B. and I.C.W. performed the behavioural studies and data analysis with supervision from L.C.P. S.R.R. and M.F.S. contributed to the proteomics experiment and analysis. A.E.G and M.A.C. performed WGS and proteome analyses. T.R. helped with patient-skin fibroblast culture and comet assay. E.J.H.vH. and U.C. performed multielectrode array data analysis with supervision from N.N.K. S.P. performed IVF. N.N. and RB performed Golgi-Cox staining with supervision from K.M.H. C.L.vE advised on RNA-Seq experiments. J.E.N. contributed to flow cytometry experiment and analysis. M.W. designed the ex37-38 del CRISPR/Cas9 editing strategy with supervision from P.Q.T. D.F. designed and helped with RT-qPCR assays. G.B. and C.P. contributed with patient clinical diagnosis and analysis. P.Q.T. supervised IVF, in vivo neurobiology assays, and behavioural testing. R.B., R.K., and J.G. wrote the original draft with input from L.A.J., M.A.C., L.C.P., and P.Q.T. All authors reviewed and edited the manuscript. R.K. and J.G. jointly contributed to overall supervision and administration of the project.

## Competing interests

The authors declare no competing interests.

## Additional information

[1]Adelaide Medical School, The University of Adelaide, Adelaide, SA 5005, Australia. [2]Robinson Research Institute, The University of Adelaide, Adelaide, SA 5005, Australia. [3]School of Biomedicine, The University of Adelaide, Adelaide, SA 5005, Australia. [4]Discipline of Anatomy and Pathology, School of Biomedicine, The University of Adelaide, Adelaide, SA 5005, Australia. [5]Proteomics, Metabolomics and MS-imaging Core Facility, South Australian Health and Medical Research Institute, and Adelaide Medical School, The University of Adelaide, Adelaide, SA 5005, Australia. [6]Department of Human Genetics, Radboudumc, Donders Institute for Brain, Cognition, and Behavior, Nijmegen 6500 HB, the Netherlands. [7]South Australian Health and Medical Research Institute, Adelaide, SA 5000, Australia. [8]School of Biomedicine, Faculty of Health and Medical Sciences, University of Adelaide and Precision Cancer Medicine Theme, Solid Tumour Program, South Australian Health and Medical Research Institute, Adelaide, SA 5000, Australia. [9]Childhood Dementia Research Group, College of Medicine and Public Health, Flinders Health & Medical Research Institute, Flinders University, Bedford Park, Adelaide, SA 5042, Australia. [10]Undiagnosed Diseases Program, Genetic Services of WA, King Edward Memorial Hospital, Subiaco, WA 6008, Australia. [11]Western Australian Register of Developmental Anomalies, King Edward Memorial Hospital, Subiaco, WA 6008, Australia. [12]Rare Care Centre, Perth Children's Hospital, Nedlands, WA 6009, Australia. [13]These authors jointly supervised this work: Raman Kumar, Jozef Gecz. ✉e-mail: jozef.gecz@adelaide.edu.au

