## [Peer Review File · Nature Communications]

Compromised transcription-mRNA export factor THOC2 causes R-loop accumulation, DNA damage and adverse neurodevelopmentREVIEWER COMMENTS

Reviewer #1 (Remarks to the Author):

Review NCOMMS-23-26757-T

The authors generated the first mouse model of THOC2-linked NDD using a short deletion of mouse THOC2 exons 37-38, a mutation relevant to human patients. They report multiple convincing cognitive and motor behavioural data which recapitulates human disease with learning, memory and sensorimotor deficits. Histopathology analysis of brains was also conducted. At molecular level, both transcriptome and proteome datasets are provided, validating the cognitive-associated behavioural phenotypes and highlighting alteration of pathways linked to transcription, splicing and cell response to DNA damage. The mRNA nuclear export investigation, an important point given the role of THOC2 in this process, is less convincing and the conclusion that it is not affected is too strong to be supported by the data. In addition, the authors show increased gamma-H2AX DNA damage in vitro in human patient fibroblast and mouse neuronal cells while THOC2-linked NDD mice have reduced neural stem cell which present altered electrophysiological properties and synapses.

Overall, this is a novel and exciting study with most experiments leading to high quality data and rigorous analysis. The reviewer is highly supportive of this work and recommend the following points to be addressed prior to acceptance for publication in Nature Communications.

Major points to be addressed:

1/ Line 81: in human and higher eukaryotes, splicing is linked to RNA nuclear export and TREX contains UAP56/DDX39B, and early subunit of the spliceosome

2/ Formatting: Figure and Table numbers are currently not labeled.

3/ Line 81: In human and higher eukaryotes, splicing is linked to mRNA nuclear export rather than to transcription via UAP56/DDX39B, an early assembly subunit of the spliceosome, and ALYREF/THOC4 which are components of TREX. On the other hand, the conserved TREX couples transcription to mRNA nuclear export in lower eukaryotes (such as in the yeast *S. cerevisiae*), which have genomes composed of a very small proportion of intron-containing genes. The text should be modified accordingly.

4/ Line 111: The study in reference 26 does not relate to Huntington's disease, Alzheimer's disease or Amyotrophic Lateral Sclerosis. It shows that cytoplasmic aggregation of fragments of proteins in these diseases (but not mutations linked to these diseases) impairs the nucleocytoplasmic transport.

5/ Line: 167: Extended Data Fig. 1b: genetics – one offspring female would be expected to have a Thoc2 Δ /+ rather than Thoc2 Δ /Y phenotype?

6/ Line 293: In Extended Fig. 3a, it is not appropriate to conclude that there is no mRNA export perturbations using an oligo-dT FISH assay which represents a gross and qualitative assessment of the nucleocytoplasmic distribution of poly-A+ RNAs. The nuclear export of a small proportion or specific subgroups of RNAs may be affected. The conclusion from the FISH is that the bulk of the mRNA nuclear export does not seem to be altered to the Thoc2 patient fibroblasts or Delta/Y mouse neurons, however the nuclear export some mRNAs may still be affected. This is an important point regarding the paper and cytoplasmic RNA-seq transcriptomes should be identified (in addition to the whole cell transcriptomes which are already provided) to investigate the genome-wide effect to the Thoc2 deletion of exons 37-38 at RNA nuclear export level. At minimum, to support the findings in Extended Fig. 3a, the expression levels of a few selected THOC2-dependent mRNA nuclear export targets should be quantified by qRT-PCR in whole-cell, nuclear and cytoplasmic fractions.

7/ Transcriptome and proteome investigations: What is the overlap between the datasets? Do the identified DEGs correlate with altered protein expression? Is this involving the same direction of changes (up- or down-regulations)? Venn diagrams should be provided and discussed. GO analysis could also be provided on the common DEGs and differentially-expressed proteins.

8/ Line 452: images in figures 7a-b do not appear to show characteristic S9.6-stained nuclear foci that are typically seen in multiple other studies. Can better images or comments be provided?

9/ Line 472: The conclusion that "THOC2 exon 37-38 deletion interferes with R-loop homeostasis and drives the increased DNA damage in the Thoc2 Δ /Y NSCs and patient fibroblasts" is too strong and should be toned down. The deletion of exons 37-38 in Thoc2 may not drive the increased DNA damage but only contribute to it or not be a direct cause of the DNA damage. In particular, the accumulation of RNA loops appear to be weaker in comparison to the level of DNA damage showed by the increased gamma-H2AX. This could also be discussed in the discussion section. In light of this, it would be preferable to remove "R-loop accumulation" in the title, keeping DNA damage which is well supported in this study.

Minor points:

1/ Line 179: indicate quantification of western blots in bar chart fig. 1c. Often the case in other figures i.e. not all panels are described.

2/ Line 195: typo for reference 28.

3/ Line 202: Extended Data Fig. 1f: text and numbers in RNA-seq read plots are too small.

4/ Line 395: typo for reference 42.

Reviewer #2 (Remarks to the Author):

This is overall a well-written paper that represents the generation and characterization of a THOC2 mouse model, the first to be developed. While at least two different human mutations resulted in embryonic lethality in mice, they were able to identify a hypomorphic allele due to an intragenic deletion that produced live pups, albeit fewer than expected. They perform functional/behavioral characterization of the mouse and then go on to further investigate cellular defects in NSCs and neurons in mouse.

The cellular mechanism is very disjointed, Thoc2 truncation > R loop accumulation > increased DNA damage > apoptosis > NSC perturbation > neuronal > migration, migration, synapse formation and electrophysiology. It might just need to be reorganized to make it more digestible. Alternatively, a figure that shows a model / summarizes the findings could be useful for the reader.

Does the truncated THOC2 show the same pattern of expression as WT THOC2 in the mice?

For the VZ reduction in thickness - did the authors look at E14.5 (or other time points) to know how early this begins?

On pg 18 / extended data Fig 6 - the finding of premature differentiation of NPCs is a major finding - consider moving to main figure. Also, this would indicate more NSC differentiating into neurons, yet the cortical plate is not as thick as the control? is it a possible migration issue?

It's interesting that mRNA transport is not affected in the mice, and in the discussion, it's suggested that fractionation prior to RNAseq may be useful to answer the question. It would be interesting to see RNAseq of patient derived neurospheres to see if the findings are similar to mice whole brain.

It appears patient samples are available, as the authors perform some parallel studies in the human cells. Why not use human samples for more of the neurodevelopment assays, such as the RNAseq, SV size, NSC differentiation?

RESPONSE TO REVIEWER COMMENTS

The authors thank the reviewers for overall positive assessment and insightful suggestions that have helped improving the manuscript. Our responses to the reviewers' comments are provided below.

Reviewer #1 (Remarks to the Author):

Review NCOMMS-23-26757-T

The authors generated the first mouse model of THOC2-linked NDD using a short deletion of mouse THOC2 exons 37-38, a mutation relevant to human patients. They report multiple convincing cognitive and motor behavioural data which recapitulates human disease with learning, memory and sensorimotor deficits. Histopathology analysis of brains was also conducted. At molecular level, both transcriptome and proteome datasets are provided, validating the cognitive-associated behavioural phenotypes and highlighting alteration of pathways linked to transcription, splicing and cell response to DNA damage. The mRNA nuclear export investigation, an important point given the role of THOC2 in this process, is less convincing and the conclusion that it is not affected is too strong to be supported by the data. In addition, the authors show increased gamma-H2AX DNA damage in vitro in human patient fibroblast and mouse neuronal cells while THOC2-linked NDD mice have reduced neural stem cell which present altered electrophysiological properties and synapses.

Overall, this is a novel and exciting study with most experiments leading to high quality data and rigorous analysis. The reviewer is highly supportive of this work and recommend the following points to be addressed prior to acceptance for publication in Nature Communications.

Major points to be addressed:

1/ Line 81: in human and higher eukaryotes, splicing is linked to RNA nuclear export and TREX contains UAP56/DDX39B, and early subunit of the spliceosome

We have now replaced this sentence.

2/ Formatting: Figure and Table numbers are currently not labeled.

We have now added labels in each figure and table.

3/ Line 81: In human and higher eukaryotes, splicing is linked to mRNA nuclear export rather than to transcription via UAP56/DDX39B, an early assembly subunit of the spliceosome, and ALYREF/THOC4 which are components of TREX. On the other hand, the conserved TREX couples transcription to mRNA nuclear export in lower eukaryotes (such as in the yeast *S. cerevisiae*), which have genomes composed of a very small proportion of intron-containing genes. The text should be modified accordingly.

We have now replaced this sentence, also in light of point 1 above.

4/ Line 111: The study in reference 26 does not relate to Huntington's disease, Alzheimer's disease or Amyotrophic Lateral Sclerosis. It shows that cytoplasmic aggregation of fragments of proteins in these diseases (but not mutations linked to these diseases) impairs the nucleocytoplasmic transport.

Yes, that's largely correct. We have now modified this part of the text. However, the Supplementary Figure S9 of Woerner et al does show co-aggregation of THOC2 protein with expanded HTT repeats in the well-studied and validated R6/2 mouse model of Huntington's disease.

5/ Line: 167: Extended Data Fig. 1b: genetics – one offspring female would be expected to have a Thoc2 Δ /+ rather than Thoc2 Δ /Y phenotype?

Great pick. Thank you. This error in Extended Data Fig. 1b has been corrected.

6/ Line 293: In Extended Fig. 3a, it is not appropriate to conclude that there is no mRNA export perturbations using an oligo-dT FISH assay which represents a gross and qualitative assessment of the nucleocytoplasmic distribution of poly-A+ RNAs. The nuclear export of a small proportion or specific subgroups of RNAs may be affected. The conclusion from the FISH is that the bulk of the mRNA nuclear export does not seem to be altered to the Thoc2 patient fibroblasts or Delta/Y mouse neurons, however the nuclear export some mRNAs may still be affected. This is an important point regarding the paper and cytoplasmic RNA-seq transcriptomes should be identified (in addition to the whole cell transcriptomes which are already provided) to investigate the genome-wide effect to the Thoc2 deletion of exons 37-38 at RNA nuclear export level. At minimum, to support the findings in Extended Fig. 3a, the expression levels of a few selected THOC2-dependent mRNA nuclear export targets should be quantified by qRT-PCR in whole-cell, nuclear and cytoplasmic fractions.

We agree with the reviewer that as oligo-dT FISH assays would only show a gross and qualitative mRNA export perturbation, our polyA+ FISH result showing no detectable bulk mRNA nuclear retention in Thoc2 ex37-38 del cells cannot be interpreted as if there is no mRNA export perturbation. We have explained this observation more clearly in the revised results (lines 294/295) and discussion (line 673-676). We did mention in our discussion that "we cannot rule out an impact on specific mRNAs" as pointed out by the reviewer. However, we agree that RNAseq of nuclear, cytoplasmic and total RNA from *Thoc2* exon 37-38 deletion neurons, NSCs, and patient fibroblasts (and possibly also iPSC derived neurons and neural stem cells) can lead to identification of specific subgroup of RNAs whose nuclear export may be impacted by Thoc2 ex37-38 del, something the authors endeavour to do in the follow-up studies. For now, we selected Thoc2 targets Klf4, Nanog and Sox2 mRNAs that show nuclear accumulation in Thoc2 knockdown mouse embryonic stem cells (Wang et al. 2013). As these targets are highly expressed in stem cells, we optimized a nuclear-cytoplasmic fractionation protocol for the mouse NSCs (that are heavily affected by *Thoc2* exon 37-38 deletion) and performed RT-qPCR (using TaqMan probes) for Klf4, Nanog and Sox2 in the total, cytoplasmic and nuclear RNA fractions of the NSCs. We observed no significant nuclear mRNA retention for these three mRNA targets (data in **Fig. R1** is provided for the reviewer), suggesting absence of obvious impact on their export by *Thoc2* exon 37-38 deletion.

Fig. R1: RT-qPCR analysis of *Nanog*, *Klf4* and *Sox2* in total (a), cytoplasmic and nuclear fractions (b) of NSCs. The relative abundance of target mRNAs is presented as Nuclear/Cytoplasmic mRNA ratio (b). NSCs were generated from two independent embryos/genotype and experiments performed in triplicates (n = 3). Data presented as mean values ± SEM.

7/ Transcriptome and proteome investigations: What is the overlap between the datasets? Do the identified DEGs correlate with altered protein expression? Is this involving the same direction of changes (up- or down-regulations)? Venn diagrams should be provided and discussed. GO analysis could also be provided on the common DEGs and differentially-expressed proteins.

The significantly dysregulated genes and proteins do not show a strong overlap (Fig. R2). However, both the DEGs and DEPs show enrichment for similar GO terms like mRNA processing, gene expression, and embryonic/preweaning lethality, among others. The reasons for such discordance between abundance of mRNAs and their cognate proteins are highly complex and have been elegantly reviewed (Liu, Beyer & Aebersold 2016). In addition, a recent study has showed that discordance between transcripts and proteins is more pronounced in brain, likely due to neural polarity (Moritz et al. 2019). Therefore, global processes like “cellular resource allocation” and “buffering of excess mRNA variation at protein levels” often attenuate the propagation of mRNA level variation to the protein level (Liu, Beyer & Aebersold 2016). Complex tissue samples composed of multiple, heterogeneous cell types also potentially contribute to the transcriptome-proteome discordance. Finally, substantial differences in the sensitivity and depth of RNA-seq and the proteome data outputs may also contribute to variation in dysregulated genes and the proteins.

Below, in Figure R2, we provide the data for the reviewer to show the overlaps.

Fig. R2: Venn diagrams showing Overlap between dysregulated genes (DEG) and dysregulated proteins (DEP) among E18.5 (a) and P10 (b) *Thoc2^{Δ/Y}* mouse brains.

8/ Line 452: images in figures 7a-b do not appear to show characteristic S9.6-stained nuclear foci that are typically seen in multiple other studies. Can better images or comments provided?

We have now repeated this experiment using a different S9.6 antibody (Sigma, MABE1095) and generated new images that show more clearly the characteristic S9.6-stained nuclear foci (Fig. 7a-b revised with new images). We also suggest we keep our original experimental data, with the other S9.6 antibody (Kerafast, AB01137-23.0), if only for comparison, as Extended Data Fig. 7a-b.

9/ Line 472: The conclusion that “THOC2 exon 37-38 deletion interferes with R-loop homeostasis and drives the increased DNA damage in the *Thoc2^{Δ/Y}* NSCs and patient fibroblasts” is too strong and should be toned down. The deletion of exons 37-38 in *Thoc2* may not drive the increased DNA damage but only contribute to it or not be a direct cause of the DNA damage. In particular, the accumulation of RNA loops appear to be weaker in comparison to the level of DNA damage showed by the increased gamma-H2AX. This could also be discussed in the discussion section.

In light of this, it would be preferable to remove “R-loop accumulation” in the title, keeping DNA damage which is well supported in this study.

Whereas we agree with the reviewer, our observation that reduced R-loop formation in RNase H1 treated *Thoc2^{Δ/Y}* NSCs reduces DNA damage (reduced comet cells) suggests increased R-loop accumulation. If THOC2 exon 37-38 del protein does not drive DNA damage, it clearly contributes to increased DNA damage. We have toned down this conclusion in line 463 of the revised manuscript.

We have deleted “R-loop accumulation” from the title of this section (line 434). However, we would like to retain R-loop accumulation in the title of the manuscript. This is because, in the manuscript title, we are not saying “R-loop accumulation” causes DNA damage but is one of the consequences of *THOC2* exon 37-38 deletion in the mouse NSCs as well as in patient primary cells, i.e., skin fibroblasts.

Minor points:

1/ Line 179: indicate quantification of western blots in bar chart fig. 1c. Often the case in other figures i.e. not all panels are described.

We have now mentioned about the quantification of western blots in Fig. 1c, 6d, 6f and extended data Fig. 1e.

2/ Line 195: typo for reference 28.

This typo has been corrected.

3/ Line 202: Extended Data Fig. 1f: text and numbers in RNA-seq read plots are too small.

We have increased the text and numbers font in the revised Extended Data Fig. 1f.

4/ Line 395: typo for reference 42.

The typo has been corrected.

Reviewer #2 (Remarks to the Author):

This is overall a well-written paper that represents the generation and characterization of a THOC2 mouse model, the first to be developed. While at least two different human mutations resulted in embryonic lethality in mice, they were able to identify a hypomorphic allele due to an intragenic deletion that produced live pups, albeit fewer than expected. They perform functional/behavioral characterization of the mouse and then go on to further investigate cellular defects in NSCs and neurons in mouse.

The cellular mechanism is very disjointed, Thoc2 truncation > R loop accumulation > increased DNA damage > apoptosis > NSC perturbation > neuronal > migration, migration, synapse formation and electrophysiology. It might just need to be reorganized to make it more digestible. Alternatively, a figure that shows a model / summarizes the findings could be useful for the reader.

This is an excellent suggestion. We have now provided a schematic diagram (new Fig. 9) summarizing the molecular, cellular, and phenotypic impact of exon 37-38 deleted truncated THOC2 protein on *Thoc2*^{ΔY} mice.

Does the truncated THOC2 show the same pattern of expression as WT THOC2 in the mice?

The truncated THOC2 protein accumulates as compared to the WT THOC2 (Fig. 1c and Extended Data Fig. 1d-e). The exon 37-38 deleted mouse *Thoc2* transcript also shows a trend of higher expression than the WT mouse (Extended Data Fig. 1f).

For the VZ reduction in thickness - did the authors look at E14.5 (or other time points) to know how early this begins?

On reviewer's suggestion, we performed immunofluorescence analysis to determine the VZ thickness in E14.5 *Thoc2^{+/-}* and *Thoc2^{Δ/-}* mouse brains and have added this new data as Fig. 5b. Similar to the E18.5 stage, the VZ in E14.5 *Thoc2^{Δ/-}* mouse brains was also reduced in size compared to *Thoc2^{+/-}* mouse brains. We have also modified the manuscript text accordingly.

On pg 18/extended data Fig 6 - the finding of premature differentiation of NPCs is a major finding - consider moving to main figure. Also, this would indicate more NSC differentiating into neurons, yet the cortical plate is not as thick as the control? is it a possible migration issue?

We have now moved the old Extended Data Fig. 6a showing premature differentiation of *Thoc2^{Δ/-}* mouse NSCs as new main Fig. 8c. As noted by the reviewer, the NSCs do differentiate into more neurons (and astrocytes). However, the reduced cortical plate thickness can be due to reduced number of NSCs to start with, and due to high rate of apoptotic cell death (Fig 6b and c). Moreover, as the *Thoc2^{Δ/-}* mouse neurons migrate to a shorter distance *in vitro* (Fig. 8b), neural migration defect can also be contributing to the formation of smaller cortical plate in *Thoc2^{Δ/-}* mouse brains.

It's interesting that mRNA transport is not affected in the mice, and in the discussion, it's suggested that fractionation prior to RNAseq may be useful to answer the question. It would be interesting to see RNAseq of patient derived neurospheres to see if the findings are similar to mice whole brain.

While we agree that these experiments will help to identify the effect of partial loss of function of THOC2 in patient neural cells as well as fibroblasts, these experiments are more involved and beyond the scope of this manuscript. Please also see reply to Reviewer 1 comment above about mRNA export effect on select mRNAs.

It appears patient samples are available, as the authors perform some parallel studies in the human cells. Why not use human samples for more of the neurodevelopment assays, such as the RNAseq, SV size, NSC differentiation?

The reviewer is correct. We have collected patient fibroblasts and recently iPSCs (both variant and CRISPR/Cas corrected WT). Given our manuscript is focussed on the *Thoc2^{Δ/-}* mouse model, in our view, these extensive experiments using human cells and stem cells are beyond the scope of this manuscript.

References

Liu, Y, Beyer, A & Aebersold, R 2016, 'On the Dependency of Cellular Protein Levels on mRNA Abundance', *Cell*, vol. 165, no. 3, Apr 21, pp. 535-550.

Moritz, CP, Muhlhaus, T, Tenzer, S, Schulenburg, T & Friauf, E 2019, 'Poor transcript-protein correlation in the brain: negatively correlating gene products reveal neuronal polarity as a potential cause', *J Neurochem*, vol. 149, no. 5, Jun, pp. 582-604.

Wang, L, Miao, YL, Zheng, X, Lackford, B, Zhou, B, Han, L, Yao, C, Ward, JM, Burkholder, A, Lipchina, I, Fargo, DC, Hochedlinger, K, Shi, Y, Williams, CJ & Hu, G 2013, 'The THO complex

regulates pluripotency gene mRNA export and controls embryonic stem cell self-renewal and somatic cell reprogramming', *Cell Stem Cell*, vol. 13, no. 6, Dec 5, pp. 676-690.

REVIEWERS' COMMENTS

Reviewer #1 (Remarks to the Author):

The authors have performed a comprehensive review of the manuscript. They have satisfactorily addressed all comments and concerns raised by the Reviewer who recommends publication in Nature Communications.

Reviewer #2 (Remarks to the Author):

The authors have addressed my concerns. Thank you.